

# Tidal variability of nutrients in a coastal coral reef system influenced by groundwater

Guizhi Wang[1,2], Shuling Wang[1], Zhangyong Wang[1], Wenping Jing[1], Yi Xu[1], Zhouling Zhang[1], Ehui Tan[1], Minhan Dai[1,2]

[1]State Key Laboratory of Marine Environmental Science, Xiamen University, Xiamen, 361102, China
[2]College of Ocean and Earth Sciences, Xiamen University, Xiamen, 361102, China

*Correspondence to*: Guizhi Wang (gzhwang@xmu.edu.cn)

**Abstract.** To investigate variations in nitrite, nitrate, phosphate and silicate in a spring-neap tide in a coral reef system influenced by groundwater discharge, we carried out a time-series observation of these nutrients and $^{228}$Ra, a tracer of groundwater discharge, in the Luhuitou fringing reef at Sanya Bay in the South China Sea. The maximum $^{228}$Ra, 45.28 dpm 100 L$^{-1}$, appeared at a low tide and the minimum, 13.98 dpm 100 L$^{-1}$, showed up during a flood tide in the spring tide. The activity of $^{228}$Ra was significantly correlated with water depth and salinity in the spring-neap tide, reflecting the tidal-pumping feature of groundwater discharge. Concentrations of all nutrients exhibited strong diurnal variations under the combined influence of mixing of groundwater and offshore water and biological uptake and release. The amplitude of the diel change reached a maximum for nitrite, nitrate, phosphate and silicate in the spring tide, 0.46 μM, 1.54 μM, 0.12 μM, and 2.68 μM, respectively. Nitrate and phosphate were negatively correlated with water depth during the spring tide, but showed no correlation during the neap tide. Nitrite was positively correlated with water depth in the spring and neap tide. They were also significantly correlated with salinity at the ebb flow of the spring tide. We quantified variations in oxidized nitrogen (NO$_x$) and phosphate contributed by biological processes based on mixing lines of these nutrients. During both the spring and neap tide biologically contributed NO$_x$ and phosphate were significantly correlated with regression slopes of 4.60 in the spring tide and 13.37 in the neap tide, similar to the composition of these nutrients in the water column, 5.43 and 14.18, respectively. This similarity indicates that the composition of nutrients in the water column of the reef system was closely related with biological processes during both tidal periods, but the biological influence appeared to be less as inferred from the less significant correlations during the spring tide when groundwater discharge was more prominent.

## 1 Introduction

Coral reefs are considered to be one of the most sensitive and stressed ecosystems occupying the coastal zone (Ban et al., 2014). Groundwater input to coral reefs was shown to be globally important and carry a significant amount of terrestrially derived nutrients to the reef systems (D'Elia et al., 1981; Paytan et al., 2006; Houk et al., 2013). Groundwater discharge is usually enriched in N relative to P with an N:P ratio higher than the Redfield ratio, 16:1 (Redfield, 1960), because of more




efficient immobilization of P than N in coastal aquifers (Slomp and Van Cappellen, 2004). Such groundwater characterized by a high N:P ratio thus could have significant impacts on coastal reef ecosystems considering that benthic marine plants are much more depleted in P with an N:P ratio of about 30:1 (Atkinson and Smith, 1983). Cuet et al. (2011) have found that the net community production in a coral-dominated fringing reef at La Réunion, France is sustained by net uptake of new

nitrogen from groundwater and net uptake of phosphate from the ocean.

Groundwater flux onto coral reefs was found to fluctuate with the tidal cycle (Lewis, 1987; Santos et al., 2010). The contribution of groundwater discharge to the nutrient budget of adjacent marine waters of coral reefs varies greatly from one site to another around the globe and at each site varies from one tidal state to another (Paytan et al., 2006). However, there is no study to reveal variations in the composition of nutrients from spring to neap tide in reef systems influenced by

10 groundwater. Then, questions are posed: a) in coral reef systems influenced by groundwater how do the abundance and composition of nutrients vary from spring to neap tide? b) what contributes to the tidal variation of nutrients in such a system?

To address these questions, this study examined the nutrient variability in a spring-neap tidal cycle in the Luhuitou fringing reef in Sanya Bay, China during a dry season. Our previous study showed that tidally-driven groundwater discharge affected the carbonate system in the Luhuitou fringing reef (Wang et al., 2014). In this reef system, groundwater discharge

played a predominant role during the spring tide and biological activities (including photosynthesis/respiration and calcification/dissolution) dominated during the neap tide in regulating diurnal variations of the carbonate parameters. Time-series observations of nutrients carried out at the same time as for the carbonate parameters in this reef system made this study possible. The naturally occurring radioactive radium isotope, $^{228}$Ra, was utilized as a tracer of groundwater discharge in this study.

## 20  2 Materials and Methods

### 2.1 Site description

Sanya Bay is a tropical bay situated at the southern tip of Hainan Island, China in the northern South China Sea under the influence of the Southeast Asian monsoon (Fig. 1a). The coastal reef time-series station CT is located at the Luhuitou fringing reef in the southeast of Sanya Bay. The Luhuitou fringing reef is a leeward coast with low wave energy (Zhang,

2001). The Holocene deposits of coral debris and biogenic carbonate sands (secondary reef) form the surfacial unconfined aquifer around the fringing reef (Zhao et al., 1983), making groundwater a diffuse source of nutrients for the reef system. Macroalgae cover about 60%, on average, of the bottom hard substrates in the Luhuitou fringing reef (Titlyanov and Titlyanova, 2013). Living scleractinian corals were observed in the lower intertidal zone and subtidal zone with coverage of 5-40% (Titlyanov and Titlyanova, 2013; Titlyanov et al., 2014; 2015). Cyanobacteria and Rhodophyta prevailed in the upper

intertidal zone, while Rhodophyta and Chlorophyta were the most abundant in the mid and lower intertidal zones (Titlyanov et al., 2014). Rhodophyta dominated the benthic macroalgal community, 54% in the upper subtidal zone (Titlyanov and





Titlyanova, 2013). The number of species in the marine flora has increased by 28% from 1990 to 2010 with a displacement of slow-growing species likely due to anthropogenic influences and coral bleaching (Titlyanov et al., 2015). The mean coral cover has decreased in the Luhuitou fringing reef from 90% in the 1960s to 12% in 2009 (Zhao et al., 2012), likely owing to a combination of regional anthropogenic impacts and climate change (Li et al., 2012).

To the north of the Luhuitou fringing reef, the Sanya River flows into Sanya Bay with an annual average discharge of 5.86 m$^3$ s$^{-1}$ (Wang et al., 2005). Investigations of nutrients, Chl $a$ and phytoplankton in the bay have been conducted seasonally for several years (Dong et al., 2010; Wu et al., 2011; Wu et al., 2012a; 2012b) and demonstrate that the inner bay is influenced by the discharge of the Sanya River with its relatively high nutrient levels, and the central and outer bay are dominated by oceanic exchange with the South China Sea (Wu et al., 2012c). The distribution of salinity in Sanya Bay

indicates that the coastal reef station CT is outside the influence of the Sanya River plume in February (Fig. 1b) (Wang et al., 2014).

## 2.1 Sampling and measurements

The setup of the sampling platform at the time-series station CT is provided in detail in Wang et al. (2014). Briefly, water was collected using a submersible pump and depth and salinity were measured with a conductivity-temperature-depth system

(Citadel, RDI Co., USA). Discrete nutrient and radium samples were taken every 3 hours during February 6-13, 2012, except on February 7-8 (with the maximum tidal range) when the samples were collected every 2 hours. A mapping cruise was conducted in Sanya Bay during February 2-3, 2012 (Fig. 1) to evaluate the influence of the Sanya River and to constrain the end-member of the offshore water. Nutrient samples for nitrate, nitrite, phosphate and silicate were collected in Sanya Bay at surface and bottom depths using 5 L Niskin bottles. Temperature and salinity were measured using a multi-parameter sonde

YSI 6600. The salinity was reported using the Practical Salinity Scale.

     Nutrient samples were filtered with 0.45 μm cellulose acetate membranes and poisoned with 1-2‰ chloroform. One filtrate was preserved at 4°C for dissolved silicate determination, and one was frozen and kept at -20°C for nitrate, nitrite, and phosphate measurements. In the laboratory, nutrients were measured with an AA3 Auto-Analyzer (Bran-Luebbe, GmbH) following the same methods in Han et al. (2012). The analytical precision was better than 1% for nitrate and nitrite, 2% for

phosphate, and 2.8% for silicate. The detection limit was 0.04 μM for nitrate and nitrite, 0.08 μM for phosphate, and 0.16 μM for silicate. Radium samples were passed through a 1 μm cartridge filter before through a MnO$_2$-impregnated acrylic fiber (Mn-fiber) column to extract dissolved radium (Rama and Moore, 1996). The Mn-fibers were leached with 1 M solutions of hydroxylamine hydrochloride and HCl to release $^{226}$Ra and $^{228}$Ra, which were then co-precipitated with BaSO$_4$ and measured in a germanium gamma detector (GCW4022, Canbera) (Moore, 1984) with an error less than 7%.



### 2.3 Statistical and interpolation method

To gain insight into factors affecting nutrients from spring to neap tide, linear regressions were conducted between water depth, salinity, and $^{228}$Ra activity, between water depth, salinity, and nutrients concentration, and between biologically contributed nutrients during the spring and neap tide. A linear curve-fitting, $y=ax+b$, was applied using least-square
minimization algorithm to find the coefficients (a, b) of the independent variable that gave the best fit between the linear equation and the data (e.g., Press et al., 1986). A significance level of 0.05 was taken. In plotting contours in Sanya Bay, Surfer 11 was utilized with kriging interpolation.

### 3 Results and discussion

### 3.1 Time-series observations of nutrients and radium at the coastal coral reef station

Time-series observations of salinity, $^{226}$Ra, and water depth at Station CT were reported in Wang et al. (2014), which demonstrated that the water depth at Station CT varied from 0.7 to 2.1 m and the salinity ranged from 33.43 to 33.67 during February 6-13, 2012. February 6, 2012 is in the middle of the lunar month, around which spring tides are expected. To separate neap tide from spring tide days, the daily variance of water depth and salinity were plotted (Fig. 2). A sharp decrease in the variance of salinity occurred on February 10, 2012 and the variance remained low (<0.001) afterwards. Thus,
two distinctive groups stood out, with one group in the period of February 6-9, 2012 having greater variance of water depth and salinity and the other in the period of February 10-13, 2012 having less variance. Therefore, we took February 6-9, 2012 as the spring tide period and February 10-13, 2012 as the neap tide period in this work.

    The concentration of nutrients varied with different patterns from spring to neap tide (Fig. 3). Nitrite varied from 0.11 to 0.71 μM during the spring tide and from 0.12 to 0.74 μM in the neap tide with the maximum diel variation of 0.46 μM
present during the spring tide (Fig. 3a). The diurnal variation was 0.24-0.46 μM during the spring tide and 0.34-0.45 μM in the neap tide. Daily peaks of nitrite usually appeared at high tides from the spring to neap tide. The concentration was positively correlated with water depth (P<0.05) during both the spring and neap tide, but the correlation was less significant during the neap tide (Fig. 4a). Nitrate and phosphate, however, showed an opposite pattern. During the spring tide, nitrate and phosphate were negatively correlated with water depth (P<0.05)(Fig. 4b,c). They reached their peak concentrations of
1.91 μM and 0.22 μM, respectively in the late afternoon and their minima of 0.37 μM and 0.10 μM, respectively at night on February 7, 2012 (Fig. 3b,c). The diurnal variation fell in the range of 0.44-1.54 μM for nitrate and 0.04-0.12 μM for phosphate. During the neap tide, the concentrations of nitrate and phosphate varied from 0.27 to 1.32 μM for nitrate and 0.084 to 0.18 μM for phosphate with less diurnal variation in the range of 0.35-0.52 μM for nitrate and 0.04-0.05 μM for phosphate. The correlation with water depth was not significant for both nutrients (P>0.15). Nitrate is the dominant species
(>50%) of oxidized nitrogen ($NO_x$) during the spring-neap tidal period except at 2 O'clock on February 12, 2012 when the



concentrations of nitrite and nitrate were almost equal. The $NO_x$:P ratio varied from 4.78 to 12.94 in the spring-neap tide (Fig. 3c). Silicate showed a trend different from either nitrite or nitrate and phosphate (Fig. 3d). It was not significantly correlated with water depth during both spring and neap tide (P>0.2). The concentration of silicate, in general, decreased from spring to neap tide. During the spring tide, the concentration of silicate fell in the range of 4.57-7.25 μM. The daily

peak concentration of silicate appeared almost at the daily lowest salinity. The diurnal variation in silicate was 1.91-2.68 μM. Around the neap tide, however, silicate ranged from 2.89 to 5.59 μM and showed less diurnal variability, 1.44-2.09 μM.

The diurnal variation in the activity of $^{228}$Ra at Station CT was 16.5-27.37 dpm 100 $L^{-1}$ (i.e., 2.75-4.56 Bq $m^{-3}$) during the spring tide, the maximum of which appeared on February 7, and 5.31-10.55 dpm 100 $L^{-1}$ around the neap tide (Fig. 3e). The maximum $^{228}$Ra, 45.28 dpm 100 $L^{-1}$, appeared at the lowest tide on February 8 during the spring tide and the minimum,

13.98 dpm 100 $L^{-1}$, showed up during the flood tide of the spring tide on February 7. The activity of $^{228}$Ra was significantly correlated with water depth in the spring-neap tidal period (P=0.002)(Fig. 5a). This pattern reflected the variation in the groundwater discharge induced by tidal pumping in this coral reef system (Wang et al., 2014), which is also observed in other coastal regions (Burnett and Dulaiova, 2003; Santos et al., 2010).

### 3.2 Distributions of nutrients in Sanya Bay

In Sanya Bay the highest concentration of nutrients appeared near the Sanya River estuary and the concentration, in general, decreased from the northeast coast, where the influence of the Sanya River plume is apparent in winter (Wang et al., 2014), to the south and west, where the South China Sea water intrudes (Fig. 6). At stations far offshore (Stations J4-5 and W3-4), the concentrations of nitrite, nitrate and phosphate were all below the detection limit and the concentration of silicate was about 4.00 μM. At other stations, the concentration of all the nutrients remained low, but was nonetheless detectable. For

example, the maximum concentration of only 0.43 μM for nitrite, 0.70 μM for nitrate, 0.18 μM for phosphate and 7.92 μM for silicate were recorded at Station P1, the station closest to the Sanya River estuary. The small islands in Sanya Bay did not show apparent influence on the nutrients in the bay since nutrients were below their detection limits or remained low around these islands (Fig. 6). The water depth at these mapping stations was no less than 5 m and the concentration of nutrients at the bottom depth differed little from that at the surface at most of these offshore stations (Table 1). This vertical distribution

confirms that the water in Sanya Bay is relatively homogenous in February (Wang et al., 2014). The $NO_x$:P ratio was less than 7 in Sanya Bay, except at Stations P2 and L6 where the $NO_x$:P ratio was around 9.

### 3.3 What affects tidal variations in nutrients at the reef station CT?

The time-series observation of salinity at Station CT suggests that more freshwater input into the reef system occurred during the ebb flow of the spring tide than during that of the neap tide (Wang et al., 2014). The distribution of salinity in Sanya Bay

(Fig. 1b) demonstrated that the Sanya River plume affected the northeast of the bay with little impact on Station CT (Wang et al., 2014). The only source of freshwater at this site in February would be groundwater discharge. The coincidence of the





daily minimum salinity with the highest activity of $^{228}$Ra during the ebb flow of the spring tide (Fig. 3e) and the significant correlation between the activity of $^{228}$Ra and salinity during the spring-neap tidal period ($P<0.0001$)(Fig. 5b) confirms that the tidally-driven groundwater discharge occurred at the coral reef station CT. Greater groundwater discharge appeared during the ebb flow in the spring tide than in the neap tide as indicated by the higher activity of $^{228}$Ra, bringing more

groundwater into the reef system.

Under the influence of tidally-driven groundwater discharge, variations in nitrite, nitrate, phosphate and silicate during the spring tide followed a tidal pattern. Inferred from the significant correlation between nutrients and water depth during the spring tide (Fig. 4), the groundwater discharge was characterized by higher nitrate and phosphate and lower nitrite than the offshore water. The daily maximum concentration of $NO_x$, phosphate, and silicate appeared in the day time at relatively low

tides, while the minimum showed up mostly at night at high tides, indicating the mixing of tidally-driven groundwater and offshore water. During the neap tide, however, $NO_x$ and phosphate showed less diurnal variations. The daily maximum concentration of $NO_x$ and phosphate appeared around the mid-night, when a flood tide appeared. This pattern reflected dominance of biological processes, consistent with the time-series observation of dissolved oxygen at this site (Wang et al., 2014). The daily minimum showed up for $NO_x$ and phosphate in the afternoon or between mid-night and dawn at high tides,

reflecting the dominance of nutrient-deplete offshore water.

Under the controls of tidally-driven groundwater discharge and biological processes, the composition of nutrients in the reef system also differed from the spring tide to the neap tide. During the spring tide when groundwater discharge played a predominant role on regulating the concentration of nutrients in the reef system, the concentration of $NO_x$ was positively correlated with the concentration of phosphate, with a regression slope of 5.43 and $R^2$ of 0.27 (Fig. 7a). The concentration of

silicate was not significantly correlated with the concentration of $NO_x$ (Fig. 7b). During the neap tide when groundwater discharge was less prominent, the correlation between the concentrations of $NO_x$ and phosphate was more significant, with a regression slope of 14.18 and $R^2$ of 0.76. The $NO_x$:P ratio was closer to the Redfield ratio than during the spring tide. The concentration of silicate showed significant correlation with the concentration of $NO_x$ in the water column, with a regression slope of 1.24 and $R^2$ of 0.58. Diatoms dominate the phytoplankton community in Sanya Bay (Zhou et al., 2009). The

elemental ratio of Si:N is 0.80±0.35 for nanoplankton and 1.20±0.37 for netplankton (Brzezinski, 1985). The similarity of the composition of silicate and $NO_x$ in the water column to the elemental ratio of diatoms implies a biological control. Unfortunately, no information is available on particular reef primary producers and sponges that may take up/release silicate in this reef system to further the discussion. The activity of $^{228}$Ra, however, was not significantly correlated with the $NO_x$:P ratio in the water column from spring to neap tide ($P>0.05$)(Fig. 5c), indicating that the composition of nutrients in the water

column was not predominantly controlled by groundwater discharge. Therefore, we propose that biological processes predominantly controled the composition of nutrients in the reef system, but the impact was less due to groundwater discharge.

**3.4 The generation and consumption of $NO_x$ and phosphate at the reef station CT**



N and P are the general limiting nutrients for the abundance of phytoplankton in coastal ecosystems (Jickells et al., 1998). To quantify the contribution of biological processes to the variations in the $NO_x$ and phosphate at Station CT, a closer look was taken at the behaviors of nitrite, nitrate and phosphate with salinity during the falling and rising phases in the spring tide. Fig. 8 shows that these nutrients behaved differently during the two phases. During the ebb flow with a faster falling speed, nitrite,

nitrate and phosphate behaved conservatively, i.e., their concentrations were significantly correlated with salinity ($R^2 \geq 0.90$, P<0.05). These conservative behaviors indicated mixing between the groundwater discharge and the offshore water. During the flood tide with a relatively slow speed, however, nitrite showed an apparent removal signal relative to the conservative mixing line while additions of nitrate and phosphate showed up. This consumption of nitrite and generation of nitrate and phosphate were due to biological processes in this period. Based on the conservative mixing lines shown in Fig. 8, we could

estimate nitrite, nitrate and phosphate owing to mixing of the offshore water and groundwater discharge using the salinity measured at Station CT ($S_{CT}$), designated as $NO_{2mix}$, $NO_{3mix}$ and $P_{mix}$.

$$NO_{2mix} = 1.3696 \times S_{CT} - 45.7520 \qquad (1),$$

$$NO_{3mix} = -1.7797 \times S_{CT} + 60.5024 \qquad (2),$$

$$P_{mix} = -0.3565 \times S_{CT} + 12.1176 \qquad (3).$$

The differences between the measured concentrations of nutrients and the nutrient concentrations resulting from mixing represented nutrients contributed by biological processes, designated as $\Delta NO_{2bio}$, $\Delta NO_{3bio}$ and $\Delta P_{bio}$,

$$\Delta NO_{2bio} = NO_{2CT} - NO_{2mix} \qquad (4),$$

$$\Delta NO_{3bio} = NO_{3CT} - NO_{3mix} \qquad (5),$$

$$\Delta P_{bio} = P_{CT} - P_{mix} \qquad (6).$$

where the subscripts 'CT' represents the measured value at Station CT. The oxidized nitrogen contributed by biological processes, $\Delta NO_{xbio}$, is the sum of $\Delta NO_{2bio}$ and $\Delta NO_{3bio}$. Positive values represent regeneration and release of nutrients in the water column and negative values reflect uptake of nutrients by marine flora (including phytoplankton and benthic flora in this system).

The nutrients contributed by biological processes showed the greatest diurnal variation in nitrate and phosphate on

February 7, 2012, which is in the spring tide, while the greatest in nitrite on February 12, 2012, which is in the neap tide (Fig. 9). Nitrite contributed by biological processes ranged from -0.15 to 0.39 µM during the spring tide and from -0.20 to 0.40 µM during the neap tide (Fig. 9a). From 6 pm on February 8 to 6 pm on February 11, 2012, biologically contributed nitrite was positive throughout the period, indicating production of nitrite. For nitrate it was produced throughout the period from 4 am on February 8 to the midnight on February 11, 2012. During the spring tide biologically contributed nitrate varied from -

0.24 to 1.25 µM and during the neap tide it fell in the range of -0.38 to 0.70 µM. Net $NO_x$ production occurred from 6 pm on February 8 to 8 am on February 12, 2012 and $\Delta NO_{xbio}$ was negative afterwards on February 12-13, 2012, indicating net



consumption (Fig. 9b). The biological contribution of phosphate had greater diurnal variations during the spring tide than during the neap tide (Fig. 9c). The greatest diel variation during the spring tide in $\Delta P_{bio}$ appeared on February 7, 2012 when $\Delta P_{bio}$ varied from -0.027 to 0.088 μM, while during the neap tide the greatest variation occurred on February 10, 2012 when $\Delta P_{bio}$ ranged from 0.009 to 0.056 μM. Net phosphate consumption occurred throughout the period of February 12-13, 2012.

The relationship between $\Delta NO_{xbio}$ and $\Delta P_{bio}$ during the spring tide differed from that during the neap tide. During the spring tide there was significant correlation between $\Delta N_{bio}$ and $\Delta P_{bio}$, with a regression slope of 4.60 and $R^2$ of 0.16 (Fig. 10). During the neap tide, however, the correlation was much more significant with a regression slope of 13.37 and $R^2$ of 0.75. The regression slope of the regression between biologically contributed $NO_{xbio}$ and phosphate was similar to that of the significant regression between $NO_{xbio}$ and phosphate in the water column, which was 5.43 during the spring tide and 14.18

during the neap tide. This similarity indicates that the composition of nutrients in the water column was closely related with biological processes during both tidal periods, but the biological effect appeared to be less during the spring tide as inferred from the less significant correlations. The net release of nutrients during the neap tide with a very Redfield-like ratio suggests that the net nutrient fluxes in this system were likely to be dominated by the uptake and remineralization of plankton/oceanic organic particles by benthic filter feeders as observed in other reefs (e.g., Ayukai, 1995; Ribes et al., 2005;

Southwell et al., 2008; Genin et al., 2009; Monismith et al., 2010). The net uptake of nitrate and phosphate was mainly made by reef primary producers. Thus, the composition of nutrients in the water column seemed be directly related with biological contributions from the spring to neap tide. The biological influence was less during the spring tide mostly likely due to groundwater discharge. This confirms our proposal that biological processes predominantly controlled the composition of nutrients in the reef system, but the impact was less due to groundwater discharge.

Successive uptake rates of $NO_x$ were approximated by the depth-integration of the biologically contributed $NO_x$ divided by the sampling time interval from the spring to neap tide. The uptake rate ranged from -9.04 to 19.07 mmol m$^{-2}$ d$^{-1}$, which compares well with the sum of nitrate and nitrite fluxes over Ningaloo Reef, a fringing reef in Australia, -24 to 15 mmol m$^{-2}$ d$^{-1}$ (Wyatt et al., 2012). It is significantly correlated with the concentration of $NO_x$ in the water column (Fig. 11), with a slope of 14.47 and $R^2$ of 0.94 (P<0.0001), indicating the mass-transfer limitation of $NO_x$ uptake. The slope (in m d$^{-1}$) falls in

the range of the typical uptake rate coefficient for dissolved inorganic nitrogen reported in Falter et al. (2004).

**3.5 Seasonal and regional extrapolations and expectations**

This study was carried out in winter. Seasonal variations are present in the river discharge as inferred from precipitation (Wang et al., 2005) and there might be increase in the groundwater discharge and associated nutrient fluxes in summer as in other coastal systems (e.g., Lewis, 1987; Costa et al., 2006; Kelly and Moran, 2012; Wang et al., 2015). However, the

relative changes in the groundwater discharge and associated nutrient fluxes would be much smaller than those of the river. The tidally-driven feature of the groundwater discharge in this reef system might make our conclusions applicable to other




seasons. But it is likely that what we observed in a dry season might be different from what would happen in a wet season due to the involvement of other forces, e.g., upwelling in summer (Wu et al., 2012a), which merits further studies.

  In relatively oligotrophic coastal systems with coral reefs, such groundwater-associated nutrient fluxes may sustain the reef community production (Cuet et al., 2011), result in increases in diversity and occurrence of algae and sponge where

5 relatively low salinity is present (Houk and Starmer, 2010), or induce the proliferation of diatom and cyanobacteria (Blanco et al., 2011). In addition, tidally-driven groundwater into nearshore ecosystems was found to be negatively correlated with seagrass habitat condition (Houk et al., 2013). Nutrients loads via groundwater discharge may affect the community structure to move towards macroalgal blooms via bottom-up control (Lapointe, 1997) and likely play a role in the displacement of slow-growing benthic flora with fast-growing species observed in Sanya Bay in the last two decades (Titlyanov et al., 2015).

10 Future changes in these fluxes, likely caused by climate change and human activities, might make the situation worse and need to be monitored in reef protection programs and be considered in assessing the environmental health of coral reef systems, especially in regions with expected higher inputs of anthropogenic nutrients into the groundwater.

### 4 Conclusions

  The variability of nutrients in a spring-neap tidal cycle in a coral reef system in winter was revealed for the first time under

15 the synergistic control of tidally-driven groundwater discharge and biological processes. The activity of $^{228}$Ra was significantly correlated with water depth and salinity, indicating tidally-driven groundwater discharge at this site. Nitrate and phosphate were negatively correlated with salinity at the ebb flow of the spring tide, indicating that groundwater discharge was enriched in nitrate and phosphate. Nitrate, phosphate and silicate in the water column showed greater diurnal variations during the spring tide than during the neap tide, while the diel change in the concentration of nitrite demonstrated no

20 consistent pattern. The nutrient composition in the water column seemed to differ between the spring tide and neap tide, but was similar to their biological uptake/release in either tidal period for oxidized nitrogen ($NO_x$) and phosphate. This similarity indicates that variations in nutrients in the water column in the reef system were mainly regulated by biological processes. However, correlations between $NO_x$ and phosphate in the water column and between biologically contributed $NO_x$ and phosphate were less significant during the spring tide when groundwater discharge was more prominent. The concentration

25 of silicate in the water column was significantly correlated with that of $NO_x$ during the neap tide, but they were not significantly correlated during the spring tide. This indicates that the composition of nutrients in the water column was also affected by tidally-driven groundwater discharge, especially during the spring tide. Therefore, biological processes predominantly controlled the composition of nutrients in the reef system, but the impact was less due to groundwater discharge.

30 The stoichiometric relationship of $NO_x$ and phosphate from the spring to neap tide in this reef system is important in understanding how biologically processes predominantly affected these nutrients variations under the influence of tidally-driven groundwater discharge. The composition of silicate and $NO_x$ during the neap tide when groundwater discharge was




less was comparable to the elemental ratio of diatoms. The release/consumption ratio of $NO_x$:P by biological processes followed a Redfield-like ratio during the neap tide, but about one third as much during the spring tide. Whether this change in the biological release/uptake ratio of $NO_x$:P is associated with a change in the community structure needs further study.

*Supplement* Time-series data are provided in Table S1.

*Author contribution* Guizhi Wang and Minhan Dai wrote the main text of the manuscript. Guizhi Wang, Shuling Wang, Zhangyong Wang, Wenping Jing, Yi Xu, and Zhouling Wang collected samples in the field and measured the parameters. Guizhi Wang analyzed the data and did the calculations. Ehui Tan drew some of the figures.

*Competing interests* The authors declare that they have no conflict of interest.

*Acknowledgments* We thank the crew of the ship *QiongLinGao* 02706 and Junde Dong for arranging local logistic support. We appreciate the constructive comments from James Falter that have greatly improved the manuscript. This study was funded by MOST (2015CB954001) and the National Natural Science Foundation of China (41576074). Professor John Hodgkiss is thanked for his assistance with English.

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

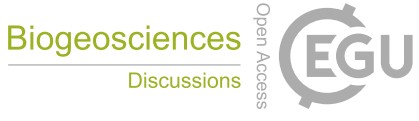



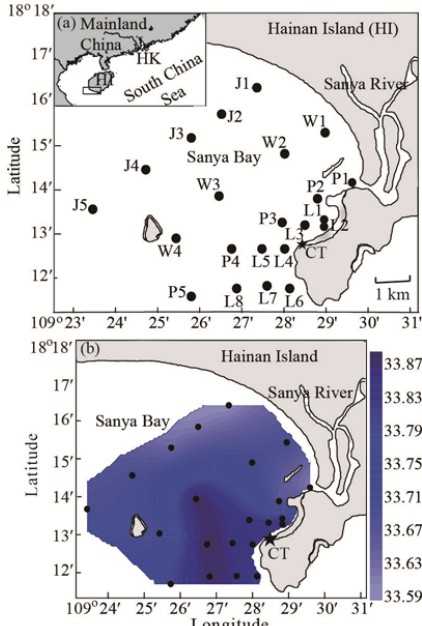

**Figure 1: Study area, sampling stations and salinity distribution in February 2012 in Sanya Bay, Hainan Island (HI) in the South China Sea. (a) study area and sampling stations; and (b) salinity distribution. HK represents Hong Kong. CT is the coastal reef time-series station.**

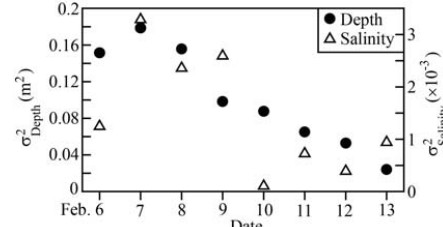

**Figure 2: Daily variance of water depth ($\sigma^2_{Depth}$) and salinity ($\sigma^2_{Salinity}$) at the coastal reef station CT during February 6-13, 2012.**





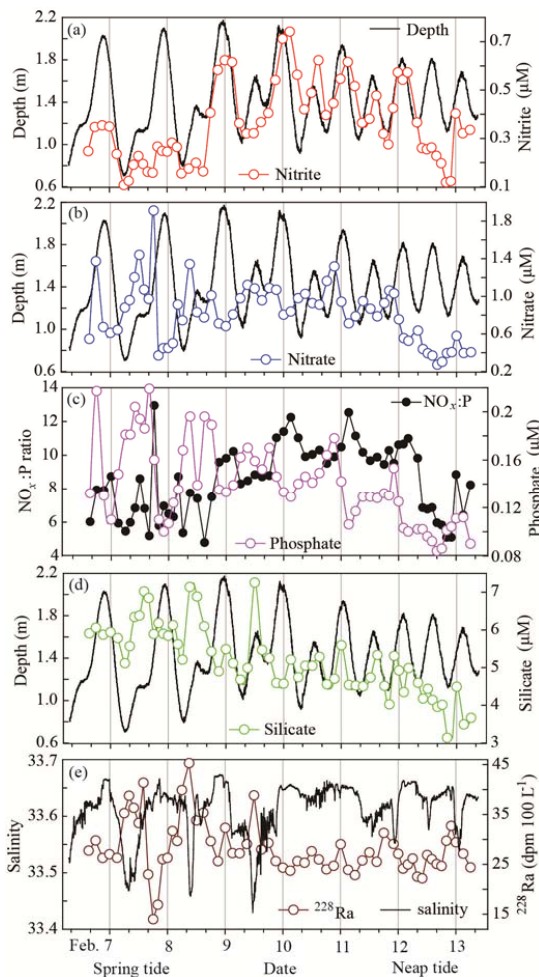

**Figure 3: Time-series observations of nutrients and [228]Ra at Station CT in the Luhuitou reef of Sanya Bay, China during February 6-13, 2012. (a) Nitrite; (b) nitrate; (c) phosphate and NO$_x$:P ratio; (d) silicate; and (e) [228]Ra. Lines connecting the symbols are to show trends. Water depth and salinity were reported in Wang et al. (2014).**





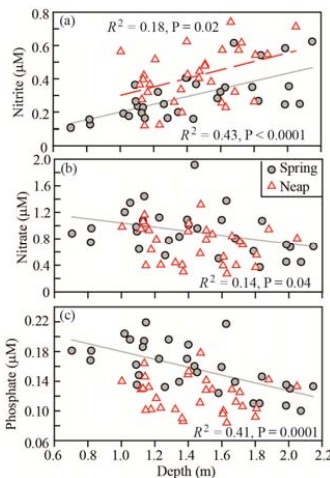

**Figure 4: Concentrations of nutrients in the water column against water depth during the spring tide and neap tide at Station CT in the Luhuitou reef during February 6-13, 2012. (a) nitrite; (b) nitrate; and (c) phosphate.**

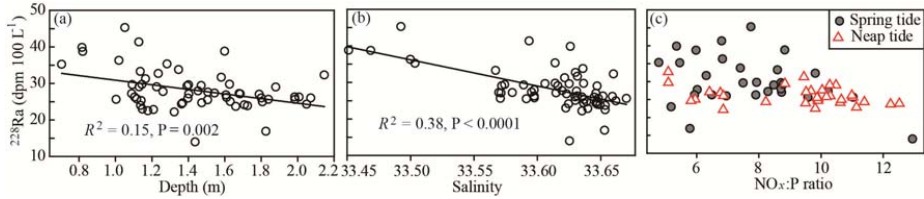

5 **Figure 5: The activity of $^{228}$Ra against water depth, salinity and the NO$_x$:P ratio in the water column at Station CT during February 6-13, 2012. (a) $^{228}$Ra vs. water depth; (b) $^{228}$Ra vs. salinity; and (b) $^{228}$Ra vs. the NO$_x$:P ratio.**




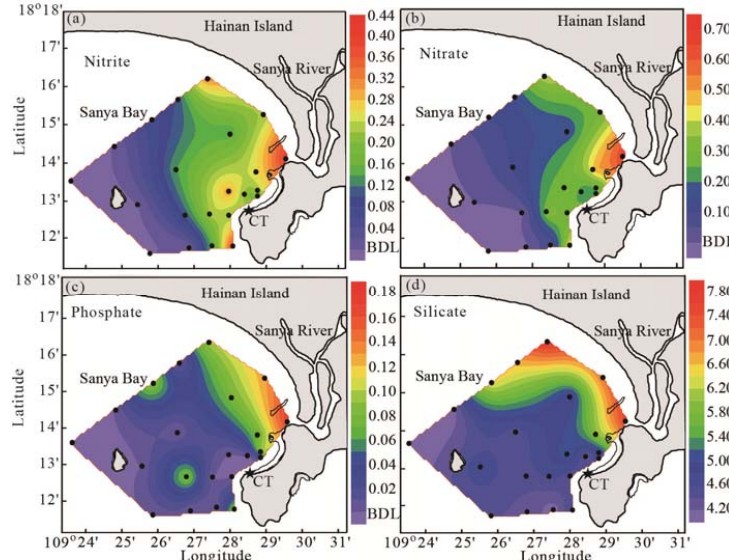

**Figure 6: Surface distributions of nutrients in Sanya Bay in February 2012. (a) Nitrite; (b) nitrate; (c) phosphate; and (d) silicate. The units are in µM. BDL is below the detection limit, which is 0.04 µM for nitrate and nitrite and 0.08 µM for phosphate.**

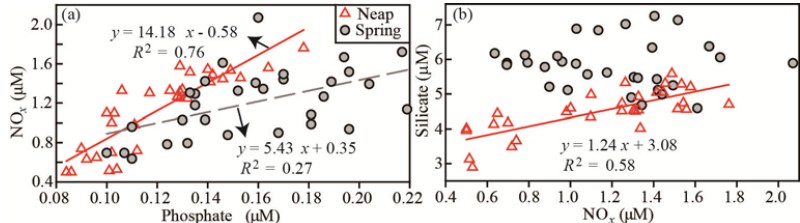

5    **Figure 7: Concentrations of nutrients in the water column against each other during the spring tide and neap tide at Station CT during February 6-13, 2012. (a) $NO_x$ against phosphate; and (b) silicate against $NO_x$.**





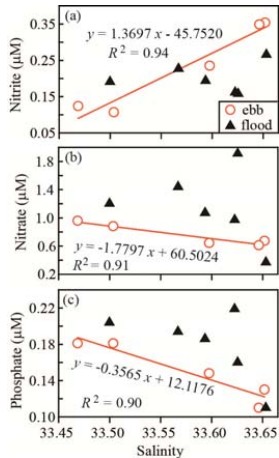

**Figure 8: Behaviours of nutrients with salinity during the ebb flow and flood tide of the spring tide at Station CT. (a) nitrite; (b) nitrate; and (c) phosphate.**

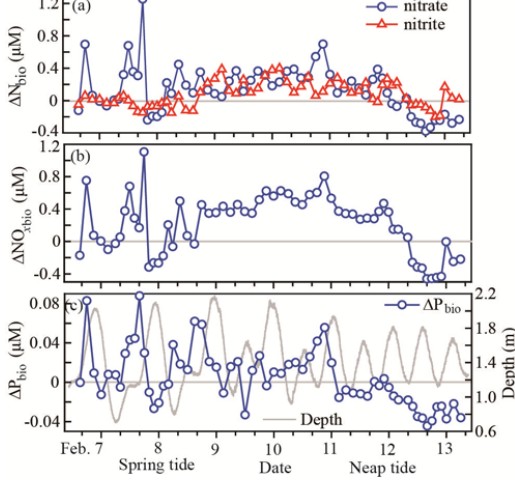

5 **Figure 9: Variations of nutrients contributed by biological processes in a spring-neap tide during February 6-13, 2012 at the coastal reef station CT. (a) nitrite and nitrate; (b) $NO_x$; and (c) phosphate (P). Water depth was reported in Wang et al. (2014).**



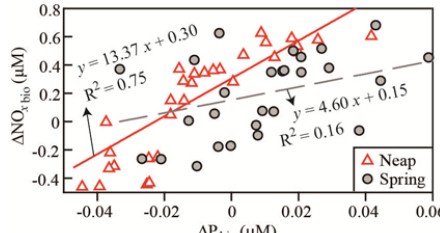

**Figure 10: Relationship between biologically contributed NO$_x$ and phosphate during the spring tide and neap tide at Station CT in the Luhuitou fringing reef in February 6-13, 2012.**

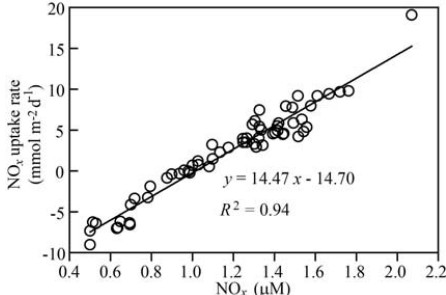

5   **Figure 11: Uptake rate of NO$_x$ against the concentration of NO$_x$ in the water column at reef Station CT in a spring-neap tide during February 6-13, 2012.**





**Table 1. Sampling stations and data collected in Sanya Bay in February 2012.**

| Station | Latitude | Longitude | Bottom Depth (m) | Sample Depth (m) | Temperature | Salinity | $NO_2^-$ (μM) | $NO_3^-$ (μM) | $PO_4^{3-}$ (μM) | $SiO_3^{2-}$ (μM) |
|---|---|---|---|---|---|---|---|---|---|---|
| J1 | 18.2718 | 109.4565 | 8 | 0.5 | 22.80 | 33.60 | 0.328 | 0.410 | 0.104 | 7.916 |
|  |  |  |  | 6.5 | 22.74 | 33.60 | 0.298 | 0.343 | 0.098 | 7.485 |
| J2 | 18.2623 | 109.4423 | 9 | 0.5 | 22.66 | 33.62 | 0.103 | 0.149 | BDL | 6.708 |
|  |  |  |  | 8.0 | 22.64 | 33.63 | 0.124 | 0.162 | BDL | 6.531 |
| J3 | 18.2531 | 109.4298 | 12 | 0.5 | 22.70 | 33.64 | 0.073 | 0.104 | 0.090 | 6.472 |
|  |  |  |  | 11.0 | 22.69 | 33.65 | 0.104 | 0.067 | 0.108 | 6.318 |
| J4 | 18.2409 | 109.4118 | 11 | 0.5 | 22.81 | 33.70 | BDL | BDL | BDL | 4.069 |
|  |  |  |  | 12.4 | 22.81 | 33.70 | BDL | BDL | BDL | 4.095 |
| J5 | 18.2261 | 109.3909 | 15 | 0.5 | 22.90 | 33.70 | BDL | BDL | BDL | 4.058 |
|  |  |  |  | 14.0 | 22.88 | 33.74 | BDL | BDL | BDL | 4.126 |
| W4 | 18.2154 | 109.4244 | 17 | 0.5 | 22.90 | 33.70 | BDL | BDL | BDL | 4.768 |
|  |  |  |  | 17.5 | 22.73 | 33.75 | BDL | BDL | BDL | 4.76 |
| W3 | 18.2306 | 109.4413 | 16 | 0.5 | 22.97 | 33.89 | 0.136 | 0.112 | BDL | 4.476 |
|  |  |  |  | 16.0 | 23.40 | 33.60 | 0.063 | 0.098 | BDL | 5.188 |
| W2 | 18.2466 | 109.4672 | 12 | 0.5 | 22.93 | 33.72 | 0.147 | 0.158 | 0.081 | 4.724 |
|  |  |  |  | 9.5 | 22.76 | 33.73 | 0.075 | 0.127 | BDL | 5.179 |
| W1 | 18.2555 | 109.4832 | 5 | 0.5 | 23.12 | 33.70 | 0.228 | 0.299 | 0.131 | 7.136 |
|  |  |  |  | 3.0 | 22.92 | 33.73 | 0.228 | 0.234 | 0.102 | 6.317 |
| P3 | 18.2213 | 109.4660 | 16 | 0.5 | 22.75 | 33.84 | 0.300 | 0.309 | BDL | 5.172 |
|  |  |  |  | 16.0 | 22.87 | 33.76 | 0.132 | 0.144 | BDL | 4.655 |
| P2 | 18.2296 | 109.4797 | 11 | 0.5 | 23.01 | 33.67 | 0.262 | 0.496 | 0.082 | 6.035 |
|  |  |  |  | 11.0 | 22.90 | 33.77 | 0.206 | 0.204 | BDL | 4.569 |
| P1 | 18.2355 | 109.4940 | 5 | 0.5 | 22.98 | 33.62 | 0.426 | 0.699 | 0.178 | 7.726 |
|  |  |  |  | 2.8 | 22.97 | 33.64 | 0.350 | 0.525 | 0.157 | 7.671 |
| P4 | 18.2105 | 109.4464 | 12 | 0.5 | 22.71 | 33.89 | 0.108 | 0.002 | 0.081 | 4.519 |
|  |  |  |  | 19.0 | 22.67 | 33.89 | 0.200 | 0.013 | 0.130 | 4.935 |
| P5 | 18.1931 | 109.4296 | 26 | 0.5 | 22.69 | 33.81 | BDL | BDL | BDL | 4.428 |
|  |  |  |  | 26.0 | 22.74 | 33.87 | 0.054 | 0.076 | BDL | 4.522 |
| L8 | 18.1964 | 109.4476 | 25 | 0.5 | 22.78 | 33.88 | BDL | BDL | BDL | 4.282 |
|  |  |  |  | 25.5 | 22.75 | 33.88 | 0.191 | 0.005 | 0.082 | 4.528 |
| L7 | 18.1966 | 109.4601 | 32 | 0.5 | 22.83 | 33.86 | 0.171 | 0.092 | BDL | 4.093 |
|  |  |  |  | 30.7 | 22.78 | 33.87 | 0.077 | 0.081 | BDL | 4.4 |
| L6 | 18.1965 | 109.4694 | 23 | 0.5 | 22.79 | 33.82 | 0.420 | 0.516 | 0.097 | 4.859 |
|  |  |  |  | 27.0 | 22.77 | 33.87 | 0.405 | 0.431 | 0.112 | 4.839 |
| L5 | 18.2111 | 109.4582 | 21 | 0.5 | 22.74 | 33.85 | 0.231 | 0.326 | BDL | 4.643 |
|  |  |  |  | 18.0 | 22.79 | 33.86 | 0.355 | 0.392 | 0.097 | 4.48 |
| L4 | 18.2105 | 109.4674 | 20 | 0.5 | 22.76 | 33.85 | 0.219 | 0.248 | BDL | 4.484 |
|  |  |  |  | 21.0 | 22.77 | 33.87 | 0.285 | 0.309 | BDL | 4.645 |
| L3 | 18.2201 | 109.4749 | 12 | 0.5 | 22.79 | 33.84 | 0.194 | 0.193 | BDL | 4.315 |
|  |  |  |  | 12.8 | 22.79 | 33.86 | 0.202 | 0.183 | BDL | 4.444 |
| L2 | 18.2193 | 109.4812 | 11 | 0.5 | 22.81 | 33.85 | 0.192 | 0.309 | BDL | 5.006 |
|  |  |  |  | 11.0 | 22.80 | 33.86 | 0.195 | 0.218 | BDL | 4.639 |
| L1 | 18.2219 | 109.4812 | 11 | 0.5 | 22.76 | 33.84 | 0.244 | 0.253 | 0.101 | 4.887 |
|  |  |  |  | 11.0 | 22.81 | 33.87 | 0.353 | 0.235 | 0.107 | 5.252 |

Note: BDL is below the detection limit.