# Peer review of "Tidal variability of nutrients in a coastal coral reef system influenced by groundwater"

_Biogeosciences, 2017_

## Referee Comment (RC1) · Anonymous Referee #1 · 1 Jun 2017

General Comments

Apart from river and surface water runoff subsurface discharge of groundwater plays a key role in coastal water and nutrient budgets. In this study, the authors discuss about nutrients and 228Ra measurements made during ebb and flood phases of spring and neap tides. Although most of the stations are in close proximity to the coastline, the authors have not reported any data from groundwater or river/stream waters for nutrients and Ra isotopes to substantiate the submarine groundwater input. Ra isotopes are also released by shelf sediments at mid-salinities. If it was measured, this will help in understanding the exchange from land to coastal bay. Some of the results are already published in the papers quoted by the authors.

Page 1:

[Figure]

Line 14: The authors claim that the diurnal variability in nutrients is due to the mixing of groundwater and offshore water and biological uptake and release. This manuscript does not show any results of biological measurements then how did the authors confirm that it is biological uptake and release during neap tide and groundwater input during spring tide?

Line 17: It is mentioned that nitrite was positively correlated with water depth in the spring and neap tides. This sentence does not convey the authors' message clearly. In general, during spring tide, seawater level (tidal height) in the bay will be high whereas during neap tide, it will be low. How can nitrite be high in both spring and neap tides in order to show positive correlation with water depth? If so, what is the mechanism for this to happen?

Line 18: The ebb flow of the spring tide would have decreased salinity and indicates the receding seawater. What is the significant correlation between nutrients and salinity? Is it is positive or negative? This should be explained here briefly and elaborated in the discussion section.

Line 19: "by biological processes based on mixing lines of these nutrients". The deviation from the mixing line need not necessarily represent biological process alone and it may be through any other addition or removal processes in the Bay.

Line 24: "less significant correlations". Quantify them.

Page 2:

Site Description:

This section lacks basic information about the study area viz. (1) the peak rainfall and runoff period of the river and what is the annual river discharge and how it affects the salinity (2) The samples were collected during which season (although it is mentioned as a dry season, in introduction section, more details should be presented in this section) and what are the river and bay conditions during the sampling season (3) Is the

river regulated by a dam in the upstream (4) Is the river fed by summer or winter monsoon (4) what is the tidal pattern and amplitude in the bay (5) Is there any tide gauge station near the study area (if so, give the location on the map) and give the tidal variations during the study period? (6) At the end of the manuscript it is explained that the region experiences upwelling (Section 3.5; page 9) but not mentioned in this section.

Line 16: (...with the maximum tidal range). Provide the tidal range with a reference.

Line 14: It is mentioned that in this reef system, groundwater play a predominant role but there is no measurement of groundwater sample. Any measurement from lake/well/river/water pump will help us to understand the concentration in the groundwater and the exchange with the bay provided with their earlier work. The diurnal variations in nutrients observed during spring and neap tides may relate to mixing reactions like release/adsorption of nutrients as well. The mixing of high saline seawater and less saline freshwater may create mixing zones with different chemical and physical properties that create changes in nutrient concentrations. This is not addressed in the paper.

Page 4:

Line 1: Statistical and Interpolation method. The sentence is not clear. Rewrite this.

Line 7: Why particularly kriging interpolation was done? Give specific reason to use this algorithm.

Results and Discussion:

This section mostly presents the results of the study without much discussion. The first 2 paragraphs explain the results and at the end of the third paragraph, there are a few references cited to just compare these results with other. Not much scientific discussion has been done to explain the reasons for such variations and for identifying processes regulating these changes. The authors should discuss Results and Discussion separately, so that readers can understand the implications of the results.

**BGD**

Section 3.1 describes nutrients and 228Ra at a time-series station followed by Section 3.2 explaining the nutrients in Sanya Bay and Section 3.3 again on the tidal variations in nutrient at reef station CT. The authors could have explained the results from the time-series station CT, the influence of tides on nutrient variability and then described on Sanya Bay.

Line 13: It is that "in the middle of the lunar month. . ..expected". If this is based on the tidal gauge data, reference to that should be made.

Page 5:

Line 29: How the authors are claiming that freshwater is more during ebb flow of spring tide? Please give supporting information and include reference.

Line 31: "The only source of freshwater at this site in February would be groundwater discharge". If so, provide reference. If there are earlier studies on turbidity maxima in the bay or the coastal/estuary of the study region, then it would help in discussing the role of suspended sediments in nutrient peaks or groundwater discharge.

Line 2: P values mentioned in the manuscript varies from <0.0001 to >0.2. These are looking unrealistic from the plots. How these values are calculated, by using standard software or by using online calculations? If so, please give reference or web-link.

Line 13: The authors repeatedly mention about biological processes but no biological data has been included. It will be more appropriate to discuss the biological observations and then using mixing or dilution line calculations to identify nutrient removal/addition process. It should also be noted that in the absence of biological information, the differences (addition/removal) observed in nitrite, nitrate and phosphate could be due to sediment re-suspension and mixing. Enough scientific evidence from literature should be provided to support the arguments.

Page 7:

[Figure]

Line 12: The equations NO2mix, NO3mix, Pmix, ΔNO2bio, ΔNO3bio, ΔPbio – there are no references cited for these calculations. If this is presented first time, mention about the assumptions involved in this type of equations.

Page 11:

In the references, Kelly and Moran, 2002 is mentioned while on page 8, this year is mentioned as 2012. This requires correction.

Page 14:

Figure 1 (a) and (b). Can these two be combined as one? The figure caption has repetition. Study area, sampling stations and salinity distribution are repeated.

Page16:

Figure 4-The R2 values shown for nitrate (0.14) and nitrite (0.18) does not imply any significant relation. Is there any particular reason for the authors to show this trend line and R2 values?

Page 16:

Figure 5-The figure caption has repetition. Rewrite it.

Page 17:

Figure 6-The information like Hainan Island, Sanya river and Sanya Bay, is given in all the images (a-d). Giving these information in anyone figure will be more appropriate.

Figure 7-Rewrite the figure caption as, Concentrations of (a) NOx against phosphate and (b) silicate against NOx during . . ...

Page 19:

Figure10-What is the significance to show a trend line with R2=0.16?

Page 20:

Table 1-Give units for latitude, longitude, temperature.

[Figure]

---

## Referee Comment (RC2) · Anonymous Referee #2 · 26 Jul 2017

The manuscript provides winter observations of dissolved nitrite, nitrate, phosphate, silicate, 228Ra, salinity, and water depth in the Luhuitou fringing reef at Sanya Bay in the South China Sea. The authors introduced that in their another paper for the same cruise (Wang et al., 2014), they concluded that: tidally-driven groundwater discharge affected the carbonate system in the Luhuitou fringing reef. In this reef system, groundwater discharge played a predominant role during the spring tide and biological activities (including photosynthesis/respiration and calcification/dissolution) dominated during the neap tide in regulating diurnal variations of the carbonate parameters. Then in this study, the authors use 228Ra as a tracer of groundwater discharge to address tidal variability of nutrients in the coral reef system influenced by groundwater. It is an interesting topic. The key point supporting this manuscript is from the previous paper:

[Figure]

The time-series observation of salinity at Station CT suggests that more freshwater input into the reef system occurred during the ebb flow of the spring tide than during that of the neap tide, and the only source of freshwater at this site would be groundwater discharge (Wang et al., 2014). I have to say that I don't read such an important paper. However, based on the present presentation, the arguments provided throughout the discussion were speculative in nature. This manuscript needs major revision.

The key point to support this manuscript is that groundwater discharge played a predominant role during the spring tide in the fringing reef. The time-series observation was carried out at station CT, which is close to the coast, all the horizontal distribution plots do not cover the site, where water may source from terrigenous surface runoff, rainfall, water exchange with adjacent water, and groundwater discharge. Do the authors indicate that the groundwater discharge comes from the seabed or the coast? In general, nutrients at station CT were vertically mixed well. Is there any relation between nutrients distribution and groundwater discharge?

The authors propose that biological processes predominantly controlled the composition of nutrients in the reef system, but the impact was less due to groundwater discharge. To quantify the contribution of biological processes to the variations in the NOx and phosphate at Station CT, they took a closer look at the behaviors of nitrite, nitrate and phosphate with salinity during the falling and rising phases in the spring tide, in which only several data points were selected for the ebb flow and flood tide of the spring tide, the difference between nitrite and nitrate (or phosphate) during the flood tide was mainly due to the two points with higher salinity, the other sources or processes may affect nutrients distribution, such as nitrate and phosphate show unusual values at salinity between 33.60-33.65. Further, the authors used the relationship derived from the several data sets to estimate the consumption and then uptake rate of NOx and phosphate. In addition, what faster or slow speed of the tide means? I don't see any data support. The statements lack logic and evidence.

As for parameter measurements, the authors used 1-2% chloroform to store nutrient

samples, and gave the detection limit of 0.04 $\mu$M for nitrate and nitrite, 0.08 $\mu$M for phosphate, and 0.16 $\mu$M for silicate. I guess these values do not include water sample pretreatment and sample storage processes. As the concentrations of nutrients were low in the investigation and the variability was also low, the authors should also provide the blanks covering filtering, storage, and measurement processes.

The authors used the daily variance of water depth and salinity to separate neap tide from spring tide days (Fig. 2). In fact, the variations of water depth and salinity were not consistent. Salinity was low on Feb 6, increased on Feb 9, but dropped down on Feb 10. In addition, daily variance of water depth was shown to have unit of m2, what daily variance of water depth means? Why the authors do not use tidal level data? Water depth observations have large uncertainties.

The authors used concentrations of nutrients against water depth to see the tidal effects. Why silicate disappeared in Fig 4? Why the concentration of silicate was not significantly correlated with the concentration of NOx during the spring tide, while the concentration of silicate showed significant correlation with the concentration of NOx during the neap tide?

The authors should pay much attention to the use of significant digit. Fig. 1b is not clear enough.

---

## Author Comment (AC1) · 25 Aug 2017

Response to the comments on "Tidal variability of nutrients in a coastal coral reef system influenced by groundwater"

Anonymous Referee #1

General Comments

Apart from river and surface water runoff subsurface discharge of groundwater plays a key role in coastal water and nutrient budgets. In this study, the authors discuss about nutrients and 228Ra measurements made during ebb and flood phases of spring and

neap tides. Although most of the stations are in close proximity to the coastline, the authors have not reported any data from groundwater or river/stream waters for nutrients and Ra isotopes to substantiate the submarine groundwater input. Ra isotopes are also released by shelf sediments at mid-salinities. If it was measured, this will help in understanding the exchange from land to coastal bay. Some of the results are already published in the papers quoted by the authors.

Response: Data from groundwater close to the time-series station and the estuary for nutrients and Ra isotopes are available to confirm the submarine groundwater input. These data are going to be presented in a manuscript under preparation focused on submarine groundwater discharge (SGD) traced by Ra, and can be shown in this manuscript to confirm SGD input. It is true that desorption of radium isotopes occurs at mid-salinities. In the case of Sanya Bay the salinity in the bay is over 33, so desorption is negligible. Desorption is usually significant in estuaries, such as the Sanya River estuary where mid-salinity occurs. Diffusion from sediments is one source of radium, but it is much smaller for 228Ra than submarine groundwater discharge based on our calculation, which will be shown in our SGD manuscript. This manuscript is a sister paper of that published in Environmental Science &Technology (Wang et al., 2014, ES&T, p. 13069-13075). The ES&T paper is focused on the carbonate system in the reef system and this manuscript is focused on the nutrients. To give a context of this manuscript, especially the hydrological conditions in the bay and the reef system, it is necessary to cite some results presented in the ES&T paper in this manuscript.

Page 1:

Line 14: The authors claim that the diurnal variability in nutrients is due to the mixing of groundwater and offshore water and biological uptake and release. This manuscript does not show any results of biological measurements then how did the authors confirm that it is biological uptake and release during neap tide and groundwater input during spring tide?

Response: this claim is based on deviations from the conservative mixing of nutrients as presented in Section 3.4. The rationale is that the nutrient concentrations are determined by physical processes, such as mixing and advection, and biological processes. Advection is negligible at the reef station. Mixing results in conservative mixing of dissolved materials. The difference between the measured concentrations and those from mixing is what is contributed by biological processes. As summarized on Line 23, "the biological influence seems to be less as inferred from the less significant correlations during the spring tide." As stated on Page 6 Line 3, " Greater groundwater discharge appeared during the ebb flow in the spring tide than in the neap tide as indicated by the higher activity of 228Ra, bringing more groundwater into the reef system." On Page 8 Line 8, "the groundwater discharge was characterized by higher nitrate and phosphate and lower nitrite than the offshore water. The daily maximum concentration of NOx, phosphate, and silicate appeared in the day time at relatively low tides, while the minimum showed up mostly at night at high tides, indicating the dominance of tidally-driven groundwater discharge." As discussed in Sections 3.3 & 3.4, the composition of nutrients during the neap tide is almost the same as that contributed by biological processes (shown in Figs. 7&10), suggesting a main role played by biological processes during the neap tide.

Line 17: It is mentioned that nitrite was positively correlated with water depth in the spring and neap tides. This sentence does not convey the authors' message clearly. In general, during spring tide, seawater level (tidal height) in the bay will be high whereas during neap tide, it will be low. How can nitrite be high in both spring and neap tides in order to show positive correlation with water depth? If so, what is the mechanism for this to happen?

Response: one correction has to be made to the reviewer's statement of high seawater level (tidal height) during spring tide and low level during neap tide: the tidal range is greater during spring tide than during neap tide, not the tidal height (seawater level). As mentioned in the earlier response, the groundwater discharge was characterized

by lower nitrite than the offshore water and was greater at low tide than at high tide. Thus, at low water depth, tidally-driven groundwater discharge is greater, so that nitrite gets lower due to more groundwater in the system. At high water depth, groundwater discharge is smaller, so that nitrite gets higher due to more offshore water. Therefore, the mixing of tidally-driven groundwater with lower nitrite and offshore water with higher nitrite results in the positive correlation of nitrite with water depth.

Line 18: The ebb flow of the spring tide would have decreased salinity and indicates the receding seawater. What is the significant correlation between nutrients and salinity? Is it is positive or negative? This should be explained here briefly and elaborated in the discussion section.

Response: the correlation between nutrients and salinity was shown in Fig. 8, all with R2 of >=0.9. Nitrite is positively correlated with salinity, while nitrate and phosphate are negatively correlated with salinity. This will be explained briefly here and elaborated in the discussion in the revision as suggested.

Line 19: "by biological processes based on mixing lines of these nutrients". The deviation from the mixing line need not necessarily represent biological process alone and it may be through any other addition or removal processes in the Bay.

Response: as stated in the earlier response that the nutrient concentrations are determined by physical processes, such as mixing and advection, and biological processes. Advection is negligible at the reef station. Mixing results in conservative mixing of dissolved materials. The difference between the measured concentrations and those from mixing is what is contributed by biological processes. This statement is based on what we know about the reef system. There is no influence of river, surface runoff, or precipitation at the reef station, which will be further clarified in the revision. Adsorption/desorption from particles might be a factor influencing the phosphate concentration, as proposed for estuaries (e.g., Froelich et al., 1982, American Journal of Science, 282, p474-511; van der Zee et al., 2007, Marine Chemistry, 106, p76-91). At the reef

station the salinity is close to the seawater (>33) and the water is clear (i.e., the total suspended matter is quite low, about 15 mg/L), which makes adsorption/desorption negligible. This will be clarified in the revision.

Line 24: "less significant correlations". Quantify them.

Response: the suggestion will be taken.

Page 2:

Site Description: This section lacks basic information about the study area viz. (1) the peak rainfall and runoff period of the river and what is the annual river discharge and how it affects the salinity (2) The samples were collected during which season (although it is mentioned as a dry season, in introduction section, more details should be presented in this section) and what are the river and bay conditions during the sampling season (3) Is the river regulated by a dam in the upstream (4) Is the river fed by summer or winter monsoon (4) what is the tidal pattern and amplitude in the bay (5) Is there any tide gauge station near the study area (if so, give the location on the map) and give the tidal variations during the study period? (6) At the end of the manuscript it is explained that the region experiences upwelling (Section 3.5; page 9) but not mentioned in this section.

Response: all of the information suggested will be provided in the revision.

Line 16: (: : :with the maximum tidal range). Provide the tidal range with a reference.

Response: the suggestion will be taken.

Line 14: It is mentioned that in this reef system, groundwater play a predominant role but there is no measurement of groundwater sample. Any measurement from lake/well/river/water pump will help us to understand the concentration in the ground-water and the exchange with the bay provided with their earlier work. The diurnal variations in nutrients observed during spring and neap tides may relate to mixing re-actions like release/adsorption of nutrients as well. The mixing of high saline seawater

and less saline freshwater may create mixing zones with different chemical and physical properties that create changes in nutrient concentrations. This is not addressed in the paper.

Response: as stated in the earlier response, groundwater data and river data will be presented in the revision. The adsorption/desorption may be important for phosphate in estuaries. At the reef station the salinity is high (>33) and TSM is quite low, which makes adsorption/desorption negligible. This will be clarified in the revision.

Page 4: Line 1: Statistical and Interpolation method. The sentence is not clear. Rewrite this.

Response: the suggestion will be taken.

Line 7: Why particularly kriging interpolation was done? Give specific reason to use this algorithm.

Response: Kringing is widely used in spatial analysis and gives the best linear unbiased prediction of the intermediate values. This reason will be provided in the revision.

Results and Discussion:

This section mostly presents the results of the study without much discussion. The first 2 paragraphs explain the results and at the end of the third paragraph, there are a few references cited to just compare these results with other. Not much scientific discussion has been done to explain the reasons for such variations and for identifying processes regulating these changes. The authors should discuss Results and Discussion separately, so that readers can understand the implications of the results. Section 3.1 describes nutrients and 228Ra at a time-series station followed by Section 3.2 explaining the nutrients in Sanya Bay and Section 3.3 again on the tidal variations in nutrient at reef station CT. The authors could have explained the results from the time-series station CT, the influence of tides on nutrient variability and then described on Sanya Bay.

Response: the suggestions will be taken in the revision.

Line 13: It is that "in the middle of the lunar month: : :.expected". If this is based on the tidal gauge data, reference to that should be made.

Response: the suggestion will be taken in the revision.

Page 5:

Line 29: How the authors are claiming that freshwater is more during ebb flow of spring tide? Please give supporting information and include reference.

Response: details supporting this claim from the cited reference here will be provided.

Line 31: "The only source of freshwater at this site in February would be groundwater discharge". If so, provide reference. If there are earlier studies on turbidity maxima in the bay or the coastal/estuary of the study region, then it would help in discussing the role of suspended sediments in nutrient peaks or groundwater discharge.

Response: the suggestions will be taken.

Line 2: P values mentioned in the manuscript varies from <0.0001 to >0.2. These are looking unrealistic from the plots. How these values are calculated, by using standard software or by using online calculations? If so, please give reference or web-link.

Response: these are calculated using the software SigmaPlot. Reference will be provided.

Line 13: The authors repeatedly mention about biological processes but no biological data has been included. It will be more appropriate to discuss the biological observations and then using mixing or dilution line calculations to identify nutrient removal/addition process. It should also be noted that in the absence of biological information, the differences (addition/removal) observed in nitrite, nitrate and phosphate

could be due to sediment re-suspension and mixing. Enough scientific evidence from literature should be provided to support the arguments.

Response: as stated in earlier responses we infer biological processes from deviations from the mixing lines. We took advantage of dissolved inorganic nutrients and radium data to infer processes affecting nutrients concentrations. We can infer biological processes by eliminating other potential source/sink terms, such as re-suspension of sediments, without biological observations. This sort of information will be provided with references to support our discussion.

Page 7: Line 12: The equations NO2mix, NO3mix, Pmix, _NO2bio, _NO3bio, _Pbio – there are no references cited for these calculations. If this is presented first time, mention about the assumptions involved in this type of equations.

Response: references will be provided.

Page 11: In the references, Kelly and Moran, 2002 is mentioned while on page 8, this year is mentioned as 2012. This requires correction.

Response: 2002 is the correct year. Correction will be made. Good catch.

Page 14:

Figure 1 (a) and (b). Can these two be combined as one? The figure caption has repetition. Study area, sampling stations and salinity distribution are repeated.

Response: these two will be combined into one figure.

Page16:

Figure 4-The R2 values shown for nitrate (0.14) and nitrite (0.18) does not imply any significant relation. Is there any particular reason for the authors to show this trend line and R2 values?

Response: The reason that the two correlations are shown is that their P values are

less than 0.05, the significance level. A small R2 just implies that the correlation is not as good as that with a greater R2. The value of R2 alone can't be used to judge whether or not a correlation is significant.

Page 16: Figure 5-The figure caption has repetition. Rewrite it.

Response: the suggestion will be taken.

Page 17: Figure 6-The information like Hainan Island, Sanya river and Sanya Bay, is given in all the images (a-d). Giving these information in anyone figure will be more appropriate.

Response: the suggestion will be taken.

Figure 7-Rewrite the figure caption as, Concentrations of (a) NOx against phosphate and (b) silicate against NOx during : : :..

Response: the suggestion will be taken.

Page 19: Figure10-What is the significance to show a trend line with R2=0.16?

Response: The P value for the linear regression is less than 0.05, so the correlation is regarded as significant and shown here. A small R2 just implies that the correlation is not as good as that with a greater R2. The value of R2 alone can't be used to judge whether or not a correlation is significant.

Page 20: Table 1-Give units for latitude, longitude, temperature.

Response: the suggestion will be taken.

Anonymous Referee #2

The manuscript provides winter observations of dissolved nitrite, nitrate, phosphate, silicate, 228Ra, salinity, and water depth in the Luhuitou fringing reef at Sanya Bay in the South China Sea. The authors introduced that in their another paper for the same cruise (Wang et al., 2014), they concluded that: tidally-driven groundwater discharge affected the carbonate system in the Luhuitou fringing reef. In this reef system, groundwater discharge played a predominant role during the spring tide and biological activities (including photosynthesis/respiration and calcification/dissolution) dominated during the neap tide in regulating diurnal variations of the carbonate parameters. Then in this study, the authors use 228Ra as a tracer of groundwater discharge to address tidal variability of nutrients in the coral reef system inifluenced by groundwater. It is an interesting topic. The key point supporting this manuscript is from the previous paper: The time-series observation of salinity at Station CT suggests that more freshwater input into the reef system occurred during the ebb flow of the spring tide than during that of the neap tide, and the only source of freshwater at this site would be groundwater discharge (Wang et al., 2014). I have to say that I don't read such an important paper. However, based on the present presentation, the arguments provided throughout the discussion were speculative in nature. This manuscript needs major revision. The key point to support this manuscript is that groundwater discharge played a predominant role during the spring tide in the fringing reef. The time-series observation was carried out at station CT, which is close to the coast, all the horizontal distribution plots do not cover the site, where water may source from terrigenous surface runoff, rainfall, water exchange with adjacent water, and groundwater discharge. Do the authors indicate that the groundwater discharge comes from the seabed or the coast? In general, nutrients at station CT were vertically mixed well. Is there any relation between nutrients distribution and groundwater discharge? The authors propose that biological processes predominantly controlled the composition of nutrients in the reef system, but the impact was less due to groundwater discharge.

Response: this manuscript is a sister of the paper published in Environmental Science &Technology (2014, p. 13069-13075). The hydrological conditions in the bay and the reef system already presented in the ES&T paper were cited in this manuscript to give the context. The ES&T paper is focused on the carbonate system in the reef system and this manuscript is focused on the nutrients. There is no surface runoff or river influence around Station CT in winter. No rainfall was observed at least one

week before our sampling. So the only possible source of fresh water at this station is groundwater. This is confirmed by the significant negative correlation between 228Ra and salinity as presented in Fig. (5b). Water exchange with the adjacent ocean water was already considered in the manuscript. In the revision horizontal distributions will be plotted to cover the time-series station. At Station CT, because it is so close to the coast, the groundwater discharge from the seabed is that from the coast. Although nutrients peaks appeared around the highest 228Ra activity (the greatest groundwater discharge), the correlation between nutrients and 228Ra is not significant.

To quantify the contribution of biological processes to the variations in the NOx and phosphate at Station CT, they took a closer look at the behaviors of nitrite, nitrate and phosphate with salinity during the falling and rising phases in the spring tide, in which only several data points were selected for the ebb flow and flood tide of the spring tide, the difference between nitrite and nitrate (or phosphate) during the flood tide was mainly due to the two points with higher salinity, the other sources or processes may affect nutrients distribution, such as nitrate and phosphate show unusual values at salinity between 33.60-33.65. Further, the authors used the relationship derived from the several data sets to estimate the consumption and then uptake rate of NOx and phosphate. In addition, what faster or slow speed of the tide means? I don't see any data support. The statements lack logic and evidence.

Response: Data during the ebb flow and the flood tide of the spring tide on Feb. 7, when the full moon occurred, were selected as shown in Fig. 8 in order to examine how mixing played a role in regulating the concentrations of nutrients. Tidal-driven SGD is most prominent during the lowest tide, which occurred at the time-series station on Feb. 7, 2012 as shown in Wang et al. (2014, ES&T). Mixing of SGD and offshore water would be most obvious from data on this day. These are the reasons why only data on this day were selected. There are 5 data points for the ebb flow of the spring tide on Feb. 7, 2012. As Fig. 8 showed, these 5 points gave a reasonable and good linear fit (i.e., there is no unusual data), which indicates mixing dominance during this period on

the concentrations of nutrients and is a good representation of the mixing relationship at this site. During the flood tide on Feb. 7, 2012, as shown by dark triangles in Fig. 8, and at all other time from the spring to neap tide deviations from the mixing line for any data point represent contributions from biological processes. The logic is clear here. From the water depth vs. date plot (Fig. 3), the tidal speed can be estimated from the difference in water depth divided by the difference in date (i.e., deltah/deltat), the slope of the curve. This will be added in the revision when mentioning faster or slow speed of the tide for clarity.

As for parameter measurements, the authors used 1-2% chloroform to store nutrient samples, and gave the detection limit of 0.04 $\mu$M for nitrate and nitrite, 0.08 $\mu$M for phosphate, and 0.16 $\mu$M for silicate. I guess these values do not include water sample pretreatment and sample storage processes. As the concentrations of nutrients were low in the investigation and the variability was also low, the authors should also provide the blanks covering filtering, storage, and measurement processes.

Response: the blanks were directly set up as the baselines during the measurement process and subtracted. This will be added here in the revision. Our lab participated in the international inter-comparison of seawater nutrients analysis in 2006 and 2008 for samples collected in the North Pacific Ocean, which concentration ranged from 0.1-42.4 mmol kg-1 for nitrate, 0.0-0.6 mmol kg-1 for nitrite, 0.0-3.0 mmol kg-1 for phosphate, organized by the Geochemical Research Department of the Meteorological Research Institute (MRI) of Japan with labs from more than 15 countries including U.S.A, Japan, U.K., Germany, France, China, and Canada. Our data compared well with the consensus mean of these samples. So our measurements are reliable.

The authors used the daily variance of water depth and salinity to separate neap tide from spring tide days (Fig. 2). In fact, the variations of water depth and salinity were not consistent. Salinity was low on Feb 6, increased on Feb 9, but dropped down on Feb 10. In addition, daily variance of water depth was shown to have unit of m2, what daily variance of water depth means? Why the authors do not use tidal level data?

Water depth observations have large uncertainties. The authors used concentrations of nutrients against water depth to see the tidal effects.

Response: We do have tidal level data. But it is kind of subjective to separate the spring tide period from the neap tide period for these continuous days. For a full-moon day and a quarter-moon day it is easy to tell them apart. So we thought about doing this separation quantitatively and came up with this variance idea. Variance is the expectation of the squared deviation of a random variable from its mean and represents how far a set of numbers are spread out from their average value (Wikipedia or any text book of statistics). Daily variance is the daily average squared deviation from the mean. So it has a unit of m2 for daily variance of water depth. To cut a line between the spring tide and neap tide, the criteria is to look for a distinct difference in the pattern of the daily variances of water depth and salinity between adjacent days during the period of the full-moon day (Feb. 7, 2012) to the quarter-moon day (Feb. 14, 2012). That is how we cut the line between Feb. 9 and Feb. 10, 2012. In the revision the formula of variance will be provided for clarity.

Why silicate disappeared in Fig 4? Why the concentration of silicate was not sig-nificantly correlated with the concentration of NOx during the spring tide, while the concentration of silicate showed significant correlation with the concentration of NOx during the neap tide?

Response: Silicate was accidently cut in Fig. 4. Thanks for catching this. It will be added back. That silicate was not significantly correlated with NOx during the spring tide, while was significantly correlated with NOx during the neap tide was because SGD was more prominent during the spring tide so that biological signals were compressed by mixing and silicate and NOx were not significantly correlated. During the neap tide SGD was less and biological processes were predominant in regulating the composition of nutrients. This is consistent with our conclusions.

The authors should pay much attention to the use of significant digit. Fig. 1b is not

clear enough.

Response: Significant digits will be checked and corrected. Fig. 1b will be plotted with higher resolution.

[Figure]

[Figure]

**Fig. 1.** Figure 2: Daily variance of water depth (2Depth) and salinity (2Salinity) at the coastal reef station CT during February 6-13, 2012.

[Figure]

**Fig. 2.** Figure 3: Time-series observations of nutrients and 228Ra at Station CT in the Luhuitou reef of Sanya Bay, China during February 6-13, 2012.

[Figure]

**Fig. 3.** igure 5: The activity of 228Ra against water depth, salinity and the NOx:P ratio in the water column at Station CT during February 6-13, 2012.

[Figure]

**Fig. 4.** Figure 8: Behaviours of nutrients with salinity during the ebb flow and flood tide of the spring tide at Station CT.

[Figure]

[Figure]

[Figure]

**Fig. 5.** Figure 7: Concentrations of nutrients in the water column against each other during the spring tide and neap tide at Station CT during February 6-13, 2012.

**Fig. 6.** Figure 10: Relationship between biologically contributed NOx and phosphate during the spring tide and neap tide at Station CT in the Luhuitou fringing reef in February 6-13, 2012.

---

## Author Response (AR1)

**Response to the comments on "Tidal variability of nutrients in a coastal coral reef system influenced by groundwater"**

Responses are in blue with page and line numbers provided where changes were made in the revision. The manuscript submitted to Journal of Marine Systems, which presents SGD-associated nutrients fluxes into Sanya Bay traced by distributions of radium isotopes, is cited in the revision (Page 3, Line 25) and enclosed.

**Anonymous Referee #1**

General Comments

Apart from river and surface water runoff subsurface discharge of groundwater plays a key role in coastal water and nutrient budgets. In this study, the authors discuss about nutrients and 228Ra measurements made during ebb and flood phases of spring and neap tides. Although most of the stations are in close proximity to the coastline, the authors have not reported any data from groundwater or river/stream waters for nutrients and Ra isotopes to substantiate the submarine groundwater input. Ra isotopes are also released by shelf sediments at mid-salinities. If it was measured, this will help in understanding the exchange from land to coastal bay. Some of the results are already published in the papers quoted by the authors.

**Response**: Nutrients and Ra Data from groundwater close to the time-series station and the Sanya River estuary are available and are presented in a manuscript focused on the contribution of submarine groundwater discharge (SGD) to the nutrients budget in Sanya Bay, which was submitted to Journal of Marine Systems (JMS). The JMS manuscript is referred to in the revision (Page 3, Line 25) and provided for the review process to substantiate the SGD input.

It is true that desorption of radium isotopes occurs when fresh water encounters seawater and Ra desorption reaches the maximum at mid-salinities. In the case of Sanya Bay the salinity in the bay is over 33, so desorption is negligible. Diffusion from sediments is one source of radium, but it is much smaller for $^{228}$Ra than submarine groundwater discharge based on our calculation as shown in our JMS manuscript.

This manuscript is a sister paper of the one published in Environmental Science &Technology (Wang et al., 2014, ES&T, p. 13069-13075). Both papers are based on the time-series observations at the coral reef station. However, the ES&T paper is focused on the carbonate system in the reef system and this manuscript is focused on the nutrients. To give a context of this manuscript, especially the hydrological conditions in the bay and the reef system, it is necessary to cite results presented in the ES&T paper in this manuscript.

Page 1:

Line 14: The authors claim that the diurnal variability in nutrients is due to the mixing of groundwater and offshore water and biological uptake and release. This manuscript does not show any results of biological measurements then how did the authors confirm that it is biological uptake and release during neap tide and groundwater input during spring tide?

**Response**: this claim is based on deviations from the conservative mixing of nutrients as presented in Section 4.2 (Page 8, Line 1-27). The rationale is that nutrient concentrations are determined by physical processes, such as mixing and advection, and biological processes. Advection is negligible at the reef station. Mixing results in conservative mixing of dissolved materials. The difference between the measured concentrations and those from mixing is what is contributed by biological processes. In the revision for clarity "under the combined influence…release" is removed, "deviations from" is added between "based on" and "mixing lines of these nutrients" (Page 1, Line 21), and at the end of the paragraph a summary sentence is added "Thus, the variability of nutrients in the coral reef system was regulated mainly by biological uptake and release in a spring-neap tide and impacted by mixing of tidally-driven groundwater and offshore seawater during spring tide." (Page 1, Line 26). As summarized on Page 1 Line 25, "the biological influence appeared to be less as inferred from the less significant correlations during the spring tide." As stated on Page 6 Line 26, " Greater groundwater discharge appeared during the ebb flow in the spring tide than in the neap tide as indicated by the higher activity of $^{228}$Ra, bringing more groundwater into the reef system." On Page 7 Line 1, "the groundwater discharge was characterized by higher nitrate and phosphate and lower nitrite than the offshore water. The daily maximum concentration of $NO_x$, phosphate, and silicate appeared in the day time at relatively low tides, while the minimum showed up mostly at night at high tides, indicating the dominance of tidally-driven groundwater discharge." As discussed in Sections 4.1 & 4.2, the composition of nutrients during the neap tide is almost the same as that contributed by biological processes (shown in Figs. 7&10), suggesting a main role played by biological processes during the neap tide.

Line 17: It is mentioned that nitrite was positively correlated with water depth in the spring and neap tides. This sentence does not convey the authors' message clearly. In general, during spring tide, seawater level (tidal height) in the bay will be high whereas during neap tide, it will be low. How can nitrite be high in both spring and neap tides in order to show positive correlation with water depth? If so, what is the mechanism for this to happen?

**Response**: one correction has to be made to the reviewer's statement of high seawater level (tidal height) during spring tide and low level during neap tide: the tidal range is greater during spring tide than during neap tide, not the tidal height (seawater level). As mentioned in the earlier response, the groundwater discharge was characterized by lower nitrite than the offshore seawater and was greater at low tide than at high tide. Thus, at low water depth, tidally-driven groundwater discharge is greater, so that nitrite gets lower due to more groundwater in the system. At high water depth, groundwater discharge is smaller, so that nitrite gets higher due to more offshore seawater. Therefore, the mixing of nitrite-lower tidally-driven groundwater and nitrite-higher offshore seawater results in the positive correlation of nitrite with water depth. An explanation is added here in the revision (Page 1 Line 17).

Line 18: The ebb flow of the spring tide would have decreased salinity and indicates the receding seawater. What is the significant correlation between nutrients and salinity? Is it is positive or negative? This should be explained here briefly and elaborated in the discussion section.

**Response**: the correlation between nutrients and salinity was shown in Fig. 8, all with $R^2$ of >=0.9 and P<0.05. Nitrite is positively correlated with salinity, while nitrate and phosphate are negatively correlated with salinity. In the revision brief explanation is provided here (Page 1 Line 17) and elaborated in the discussion (Page 8 Line 6) as suggested.

Line 19: "by biological processes based on mixing lines of these nutrients". The deviation from the mixing line need not necessarily represent biological process alone and it may be through any other addition or removal processes in the Bay.

**Response**: as stated in the earlier response that nutrient concentrations are determined by physical processes, such as mixing and advection, and biological processes. Advection is negligible at the reef station. Mixing results in conservative mixing of dissolved materials. The difference between the measured concentrations and those from mixing is what is contributed by biological processes. This statement is based on what we know about the reef system. There is no influence of river, surface runoff, or wet precipitation at the reef station during the two weeks before the sampling period and during the sampling period, which is further clarified in the revision (Page 6 Line 20). Adsorption/desorption from particles might be a factor influencing the phosphate concentration, as proposed for estuaries (e.g., Froelich et al., 1982, American Journal of Science, 282, p474-511; van der Zee et al., 2007, Marine Chemistry, 106, p76-91). At the reef station the salinity is close to the seawater (>33) and the water is clear (i.e., the total suspended matter is quite low, about 15 mg/L), which makes adsorption/desorption negligible. This statement is added in the discussion in the revision (Page 7 Line 8). Benthic release due to remineralization of organic matter is included in the biological processes. This clarification is also added in the discussion in the revision (Page 8, Line 29).

Line 24: "less significant correlations". Quantify them.

**Response**: the suggestion is taken (Page 1, Line 22).

Page 2:
Site Description:
This section lacks basic information about the study area viz. (1) the peak rainfall and runoff period of the river and what is the annual river discharge and how it affects the salinity (2) The samples were collected during which season (although it is mentioned as a dry season, in introduction section, more details should be presented in this section) and what are the river and bay conditions during the sampling season (3) Is the river regulated by a dam in the upstream (4) Is the river fed by summer or winter monsoon (4) what is the tidal pattern and amplitude in the bay (5) Is there any tide gauge station near the study area (if so, give the location on the map) and give the tidal variations during the study period? (6) At the end of the manuscript it is explained that the region experiences upwelling (Section 3.5; page 9) but not mentioned in this section.

**Response**: there is no tide gauge station near the study area. But tidal information from the literature was provided in the revision (Page 3 Line 4). Tidal variations based on our observations at the time-series station are demonstrated in the manuscript (Page 3 Line 32 & Page 4 Line 28). All the other information suggested is provided in the revision (Page 2 Line 28-31 to Page 3 Line 6 & Page 3 Line 19-21).

Line 16: (: : :with the maximum tidal range). Provide the tidal range with a reference.

**Response**: the suggestion is taken in the revision (Page 3 Line 32).

Line 14: It is mentioned that in this reef system, groundwater play a predominant role but there is no measurement of groundwater sample. Any measurement from lake/well/river/water pump will help us to understand the concentration in the groundwater and the exchange with the bay provided with their earlier work. The diurnal variations in nutrients observed during spring and neap tides may relate to mixing reactions like release/adsorption of nutrients as well. The mixing of high saline seawater and less saline freshwater may create mixing zones with different chemical and physical properties that create changes in nutrient concentrations. This is not addressed in the paper.

**Response**: as stated in the earlier response, groundwater data and river data are presented in the manuscript submitted to JMS, which is cited in the revision (Page 3 Line 25) to demonstrate the influence of groundwater-carried nutrients on the bay. The adsorption/desorption may be important for phosphate in estuaries. At the reef station the salinity is high (>33) and TSM is quite low, which makes adsorption/desorption negligible. This is clarified in the revision (Page 7 Line 8). The physical mixing of seawater and groundwater results in conservative behavior in nutrients as we demonstrated in the main text (Page 8 Line 5) and deviations from the mixing lines are changes due to chemical reactions, mainly caused by biological processes as we stated in our main text (Page 8 Line 10-11, 20-29).

Page 4:
Line 1: Statistical and Interpolation method. The sentence is not clear. Rewrite this.

**Response**: the suggestion is taken in the revision (Page 4 Line 16).

Line 7: Why particularly kriging interpolation was done? Give specific reason to use this algorithm.

**Response**: Kringing is widely used in spatial analysis and gives the best linear unbiased prediction of the intermediate values. This reason is provided in the revision (Page 4 Line 23).

Results and Discussion:
This section mostly presents the results of the study without much discussion. The first 2 paragraphs explain the results and at the end of the third paragraph, there are a few references cited to just compare these results with other. Not much scientific discussion has been done to explain the reasons for such variations and for identifying processes regulating these changes. The authors should discuss Results and Discussion separately, so that readers can understand the implications of the results. Section 3.1 describes nutrients and 228Ra at a time-series station followed by Section 3.2 explaining the nutrients in Sanya Bay and Section 3.3 again on the tidal variations in nutrient at reef station CT. The authors could have explained the results from the time-series station CT, the influence of tides on nutrient variability and then described on Sanya Bay.

**Response**: in the revision Results and Discussion are separated. In Results two sections are included: Section 3.1 describes variations in nutrients and $^{228}$Ra at the time-series station CT and Section 3.2 describes distributions of nutrients in Sanya Bay. In Discussion two sections are included: in Section 4.1 processes regulating these variations are identified and in Section 4.2 seasonal and regional extrapolations are discussed. (see Results and Discussion).

Line 13: It is that "in the middle of the lunar month: : :.expected". If this is based on the tidal gauge data, reference to that should be made.

**Response**: the reference with the tidal data is added in the revision (Page 4 Line 29).

Page 5:
Line 29: How the authors are claiming that freshwater is more during ebb flow of spring tide? Please give supporting information and include reference.

**Response**: details supporting this claim from the cited reference here are provided in the revision (Page 6 Line 19).

Line 31: "The only source of freshwater at this site in February would be groundwater discharge". If so, provide reference. If there are earlier studies on turbidity maxima in the bay or the coastal/estuary of the study region, then it would help in discussing the role of suspended sediments in nutrient peaks or groundwater discharge.

**Response**: the suggestion is taken (Page 6 Line 21). The concentration of total suspended matter in the area is provided in the revision to help in discussing the role of sorption/desorption (Page 7 Line 10).

Line 2: P values mentioned in the manuscript varies from <0.0001 to >0.2. These are looking unrealistic from the plots. How these values are calculated, by using standard software or by using online calculations? If so, please give reference or web-link.

**Response**: these are calculated using SigmaPlot. Reference is provided in the revision (Page 4 Line 21).

Line 13: The authors repeatedly mention about biological processes but no biological data has been included. It will be more appropriate to discuss the biological observations and then using mixing or dilution line calculations to identify nutrient removal/addition process. It should also be noted that in the absence of biological information, the differences (addition/removal) observed in nitrite, nitrate and phosphate could be due to sediment re-suspension and mixing. Enough scientific evidence from literature should be provided to support the arguments.

**Response**: as stated in earlier responses we infer biological processes from deviations from the mixing lines. We took advantage of dissolved inorganic nutrients and radium data to infer processes affecting nutrients concentrations. We can infer biological processes by eliminating other potential source/sink terms, such as sorption/desorption and re-suspension of sediments, without biological observations. This sort of information is provided with references to support our discussion in the revision (Page 7 Line 8-13). Benthic release due to remineralization of organic matter is included in the biological processes as we clarify in the revision (Page 8 Line 29).

Page 7:
Line 12: The equations NO2mix, NO3mix, Pmix, _NO2bio, _NO3bio, _Pbio – there are no references cited for these calculations. If this is presented first time, mention about the assumptions involved in this type of equations.

**Response**: assumptions are provided in the revision (Page 8 Line 18).

Page 11: In the references, Kelly and Moran, 2002 is mentioned while on page 8, this year is mentioned as 2012. This requires correction.

**Response**: 2002 is the correct year. Correction is made in the revision (Page 10 Line 6).

Page 14:
Figure 1 (a) and (b). Can these two be combined as one? The figure caption has repetition. Study area, sampling stations and salinity distribution are repeated.

**Response**: these two are combined into one figure (see Figure 1 on Page 16).

Page16:
Figure 4-The R2 values shown for nitrate (0.14) and nitrite (0.18) does not imply any significant relation. Is there any particular reason for the authors to show this trend line and R2 values?

**Response**: The reason that the two correlations are shown is that their P values are less than 0.05, the significance level. A small $R^2$ just implies that the correlation is not as good as that with a greater $R^2$. The value of $R^2$ alone can't be used to judge whether or not a correlation is significant.

Page 16:
Figure 5-The figure caption has repetition. Rewrite it.

**Response**: the suggestion is taken in the revision (see Figure 5 on Page 18).

Page 17:
Figure 6-The information like Hainan Island, Sanya river and Sanya Bay, is given in all the images (a-d). Giving these information in anyone figure will be more appropriate.

**Response**: the suggestion is taken in the revision (see Figure 6 on Page 19).

Figure 7-Rewrite the figure caption as, Concentrations of (a) NOx against phosphate and (b) silicate against NOx during : : :..

**Response**: the suggestion is taken in the revision (see Figure 7 on Page 19).

Page 19:
Figure10-What is the significance to show a trend line with R2=0.16?

**Response**: The P value for the linear regression is less than 0.05, so the correlation is regarded as significant and shown here. A small $R^2$ just implies that the correlation is not as good as that with a greater $R^2$. The value of $R^2$ alone can't be used to judge whether or not a correlation is significant.

Page 20:
Table 1-Give units for latitude, longitude, temperature.

**Response**: the suggestion is taken in the revision (see Table 1 on Page 22).

Anonymous Referee #2
The manuscript provides winter observations of dissolved nitrite, nitrate, phosphate, silicate, 228Ra, salinity, and water depth in the Luhuitou fringing reef at Sanya Bay in the South China Sea. The authors introduced that in their another paper for the same cruise (Wang et al., 2014), they concluded that: tidally-driven groundwater discharge affected the carbonate system in the Luhuitou fringing reef. In this reef system, groundwater discharge played a predominant role during the spring tide and biological activities (including photosynthesis/respiration and calcification/dissolution) dominated during the neap tide in regulating diurnal variations of the carbonate parameters. Then in this study, the authors use

228Ra as a tracer of groundwater discharge to address tidal variability of nutrients in the coral reef system influenced by groundwater. It is an interesting topic. The key point supporting this manuscript is from the previous paper: The time-series observation of salinity at Station CT suggests that more freshwater input into the reef system occurred during the ebb flow of the spring tide than during that of the neap tide, and the only source of freshwater at this site would be groundwater discharge (Wang et al., 2014). I have to say that I don't read such an important paper. However, based on the present presentation, the arguments provided throughout the discussion were speculative in nature. This manuscript needs major revision. The key point to support this manuscript is that groundwater discharge played a predominant role during the spring tide in the fringing reef. The time-series observation was carried out at station CT, which is close to the coast, all the horizontal distribution plots do not cover the site, where water may source from terrigenous surface runoff, rainfall, water exchange with adjacent water, and groundwater discharge. Do the authors indicate that the groundwater discharge comes from the seabed or the coast? In general, nutrients at station CT were vertically mixed well. Is there any relation between nutrients distribution and groundwater discharge? The authors propose that biological processes predominantly controlled the composition of nutrients in the reef system, but the impact was less due to groundwater discharge.

**Response**: this manuscript is a sister of the paper published in Environmental Science &Technology (2014, p. 13069-13075). The hydrological conditions in the bay and the reef system were presented in the ES&T paper, which was cited in this manuscript to give the context. The ES&T paper is focused on the carbonate system in the reef system and this manuscript is focused on the nutrients. There is no surface runoff or river influence around Station CT in winter. No rainfall was observed two weeks before our sampling. In the revision information of rainfall, surface runoff, and river influence are provided (Page 2 Line 28-31 to Page 3 Line 6 & Page 3 Line 19-21). So the only possible source of fresh water at this station is groundwater. This is confirmed by the significant negative correlation between $^{228}$Ra (the groundwater tracer) and salinity as presented in Fig. 5b (Page 18). Water exchange with the adjacent ocean water was already considered in the manuscript. At Station CT, which is about 30 m away from the coast with water depth of 0.7-2.1 m, the groundwater discharge is from both the seabed and the coast. Although nutrients peaks appeared around the highest $^{228}$Ra activity (the greatest groundwater discharge), the correlation between nutrients and $^{228}$Ra is not significant. This further supports the predominance of biological processes on the nutrient composition. During the spring tide, the groundwater discharge was greater than during the neap tide, and there was less significant correlation between nutrients than during the neap tide or no correlation (Figure 7, Page 19). So we propose that the impact of biological processes on the nutrient composition was less due to groundwater discharge. This is stated in the Abstract (Page 1 Line 25-29) and elaborated in Discussion (Page 7 Line 15-30 & Page 9 Line 14-28).

To quantify the contribution of biological processes to the variations in the NOx and phosphate at Station CT, they took a closer look at the behaviors of nitrite, nitrate and phosphate with salinity during the falling and rising phases in the spring tide, in which only several data points were selected for the ebb flow and flood tide of the spring tide, the difference between nitrite and nitrate (or phosphate) during the flood tide was mainly due to the two points with higher salinity, the other sources or processes may affect nutrients distribution, such as nitrate and phosphate show unusual values at salinity between 33.60-33.65. Further, the authors used the relationship derived from the several data sets to estimate the consumption and then uptake rate of NOx and phosphate. In addition, what faster or slow speed of the tide means? I don't see any data support. The statements lack logic and evidence.

**Response**: Data during the ebb flow and the flood tide on Feb. 7, when the greatest tidal range occurred during the spring tide period, was selected as shown in Figure 8 (Page 20) in order to examine how mixing played a role in regulating the concentrations of nutrients. Tidal-driven SGD is most prominent during the lowest tide, which occurred at the time-series station on Feb. 7, 2012 as shown in Wang et al. (2014, ES&T). Mixing of SGD and offshore seawater would be most obvious from data on this day. These are the reasons why only data on this day was selected. There are 5 data points for the ebb flow of the spring tide on Feb. 7, 2012. As Figure 8 showed, these 5 points gave a reasonable and good linear fit (i.e., there is no unusual data), which indicates mixing dominance during this period on the concentrations of nutrients and is a good representation of the mixing relationship at this site. During the flood tide on Feb. 7, 2012, as shown by dark triangles in Figure 8, and at all other time from the spring to neap tide deviations from the mixing line for any data point represent contributions from biological processes when other sources were eliminated such as adsorption/desorption and sediment re-suspension at this site (Page 7 Line 8-13). The logic is clear here. Two assumptions were made before setting up the mixing equations, (a) there was no other water mass into the reef system besides offshore seawater and groundwater, and (b) mixing of offshore seawater and groundwater from spring to neap tide follows the same relation derived from data on the day with the greatest tidal range. The assumptions are added in the revision (Page 8 Line 18). In the two weeks before our sampling and during our sampling period there were no rainfall and consequent surface runoff in this area, so there is no other water source with less salinity into the reef system. The only source of freshwater at this site is groundwater. This argument is also added in the revision (Page 6 Line 19-23). From the water depth vs. date plot (Figure 3, Page 17), the tidal speed can be estimated from the difference in water depth divided by the difference in time (i.e., $\Delta h/\Delta t$), the slope of the curve. This is added in the revision when mentioning fast or slow speed of the tide (Page 8 Line 5 & 9).

As for parameter measurements, the authors used 1-2% chloroform to store nutrient samples, and gave the detection limit of 0.04 μM for nitrate and nitrite, 0.08 μM for phosphate, and 0.16 μM for silicate. I guess these values do not include water sample pretreatment and sample storage processes. As the concentrations of nutrients were low in the investigation and the variability was also low, the authors should also provide the blanks covering filtering, storage, and measurement processes.

**Response**: the blanks were directly set up as the baselines during the measurement process and subtracted. This is added here in the revision (Page 4 Line 11). Our lab participated in the international inter-comparison of seawater nutrients analysis in 2006 and 2008 for samples collected in the North Pacific Ocean, which concentration ranged from 0.1-42.4 µmol kg$^{-1}$ for nitrate, 0.0-0.6 µmol kg$^{-1}$ for nitrite, 0.0-3.0 µmol kg$^{-1}$ for phosphate, and 1.7-156.1 µmol kg$^{-1}$ for silicate, organized by the Geochemical Research Department of the Meteorological Research Institute (MRI) of Japan with labs from more than 15 countries, including U.S.A, Japan, U.K., Germany, France, China, and Canada. Our data compared well with the consensus mean of these samples. So we have confidence in data.

The authors used the daily variance of water depth and salinity to separate neap tide from spring tide days (Fig. 2). In fact, the variations of water depth and salinity were not consistent. Salinity was low on Feb 6, increased on Feb 9, but dropped down on Feb 10. In addition, daily variance of water depth was shown to have unit of m2, what daily variance of water depth means? Why the authors do not use tidal level data? Water depth observations have large uncertainties. The authors used concentrations of nutrients against water depth to see the tidal effects.

**Response**: There is no tidal gauge station around this area. Data of water depth collected on a mooring buoy is good enough to resemble tidal level data. It is easy to tell the days with the greatest tidal range and the smallest tidal range in a spring-neap tide. But it is kind of subjective to separate the spring tide period from the neap tide period for these continuous days. So we thought about doing this separation quantitatively and came up with this variance idea. Variance is the expectation of the squared deviation of a random variable from its mean and represents how far a set of numbers are spread out from their average value (Wikipedia or any text book of statistics). Daily variance is the daily average squared deviation from the mean. So it has a unit of m$^2$ for daily variance of water depth. To cut a line between the spring tide and neap tide, the criteria is to look for a distinct difference in the pattern of the daily variances of water depth and salinity between adjacent days during the observation period, Feb. 6, 2012 to Feb. 13, 2012. That is how we cut the line between Feb. 9 and Feb. 10, 2012. In the revision the formula of variance is provided for clarity (Page 5 Line 2-4).

Why silicate disappeared in Fig 4? Why the concentration of silicate was not significantly correlated with the concentration of NOx during the spring tide, while the concentration of silicate showed significant correlation with the concentration of NOx during the neap tide?

**Response**: Silicate was accidently left out in Figure 4. In the revision silicate is added in Figure 4 (see Figure 4, Page 18). Silicate was not significantly correlated with NOx during the spring tide, while was significantly correlated with NOx during the neap tide because SGD was more prominent during the spring tide so that biological signals were compressed by mixing and silicate and NOx were not significantly correlated. During the neap tide SGD was less and the impact of biological processes was greater in regulating the composition of nutrients and consequently a significant correlation showed up. This is consistent with our conclusions.

The authors should pay much attention to the use of significant digit. Fig. 1b is not clear enough.

**Response**: Significant digits are checked and corrected in Tables 1 (Page 22) and s1. Figures 1a and 1b are combined into one Figure (see Figure 1, Page 16).

[revised manuscript text omitted]

**Abstract**

To quantify the contribution of submarine groundwater discharge (SGD) to the nutrients budget in tropical embayments, naturally occurring radium isotopes ($^{223}$Ra, $^{224}$Ra, $^{226}$Ra, and $^{228}$Ra) were investigated as SGD tracers in Sanya Bay, China in the northern South China Sea. Higher activities of radium were present along the north coast and near the Sanya River estuary. Using the activity ratio of $^{224}$Ra/$^{228}$Ra, the apparent water age in Sanya Bay was estimated to be 0-13.2 days, with an average of 7.2±3.2 days. Based on the mass balance of $^{226}$Ra and $^{228}$Ra, SGD was calculated to be 2.76-5.03×10$^6$ m$^3$ d-1 (or 4.3-7.7 cm d-1), which accounted for more than half of the respective radium source flux into Sanya Bay. SGD associated dissolved inorganic nutrient fluxes into Sanya Bay were estimated to be 3.91-7.11×10$^5$ mol NO$_x$ d-1, 5.03-9.15×10$^5$ mol P d-1, and 6.55-11.9×10$^5$ mol Si d-1. The estuarine nutrients flux from the Sanya River was equivalent to the phosphate flux via SGD, but a few times smaller the nitrogen and silicate fluxes carried by SGD. SGD was also more important than atmospheric deposition and nitrogen fixation in the nutrients budget. Our results demonstrate that SGD contributed at least 38% phosphate, 90% nitrogen, and 83% silicate in Sanya Bay. SGD could thus supply almost all nitrogen and silicate required by phytoplankton growth in the bay.

| | |
|---|---|
| **Keywords** | submarine groundwater discharge; radium isotopes; residence time; nutrients; China, Hainan Island, Sanya Bay |
| **Corresponding Author** | Guizhi Wang |
| **Corresponding Author's Institution** | Xiamen University |
| **Order of Authors** | Guizhi Wang, Shuling Wang, Zhangyong Wang |
| **Suggested reviewers** | Nils Moosdorf, Jinzhou Du, Virginie Sanial, Adina Paytan |

**Submission Files Included in this PDF**

**File Name  [File Type]**

Cover_letter_JMS.doc  [Cover Letter]

Highlights.docx  [Highlights]

graphic_ab.tif  [Graphical Abstract]

SGD_Sanya_JMS.docx  [Manuscript File]

To view all the submission files, including those not included in the PDF, click on the manuscript title on your EVISE Homepage, then click 'Download zip file'.

**Research Data Related to this Submission**

There are no linked research data sets for this submission. The following reason is given:
Data will be made available on request

Dear Editor,

My co-authors and I would like to submit the manuscript entitled "Significance of submarine groundwater discharge in nutrients budget in tropical Sanya Bay, China " for consideration as a research article in *Journal of Marine Systems*.

This study revealed the significance of submarine groundwater discharge (SGD) in the nutrients budget in Sanya Bay, China in the dry season using radium isotopes as SGD tracers. From our results SGD contributes at least 90% nitrogen, 83% silicate, and 38% phosphate in Sanya Bay. SGD is more important than the estuarine export from the Sanya River, the atmospheric deposition and nitrogen fixation. SGD would satisfy almost all requirements of nitrogen and silicate by phytoplankton growth in the bay. We believe that our study would be of interest to broad readers of *JMS*.

The manuscript has not been previously published, in whole or in part, and it is not under consideration by any other journal. All authors have seen the manuscript and approved the submission to your journal.

For contributions of authors: Guizhi Wang wrote the main text of the manuscript. Guizhi Wang, Shuling Wang, and Zhangyong Wang collected samples in the field and measured the parameters. Guizhi Wang analyzed the data and did the calculations.

Thank you very much in advance for considering our manuscript for potential publication at *JMS*.

Sincerely yours,

Guizhi Wang
Corresponding Author

Highlights

- Radium isotopes were used to trace SGD in Sanya Bay, China
- Nutrients flux via SGD was equivalent to or more than the estuarine export flux
- SGD is a major source of N and Si and contributes at least 38% P in Sanya Bay
- SGD could satisfy almost all requirements of N and Si by phytoplankton growth

[Figure]

Nutrients budget in Sanya Bay (Unit: mol d$^{-1}$)

**Significance of submarine groundwater discharge in nutrients budget**

**in tropical Sanya Bay, China**

Guizhi Wang[1,2]*, Shuling Wang[1], Zhangyong Wang[1]

[1]State Key Laboratory of Marine Environmental Science, Xiamen University,

Xiamen, 361102, China

[2]College of Ocean and Earth Sciences, Xiamen University, Xiamen, 361102, China

*Corresponding author: Guizhi Wang, email: gzhwang@xmu.edu.cn

Abstract

To quantify the contribution of submarine groundwater discharge (SGD) to the nutrients budget in tropical embayments, naturally occurring radium isotopes ($^{223}$Ra,

$^{224}$Ra, $^{226}$Ra, and $^{228}$Ra) were investigated as SGD tracers in Sanya Bay, China in the northern South China Sea. Higher activities of radium were present along the north coast and near the Sanya River estuary. Using the activity ratio of $^{224}$Ra/$^{228}$Ra, the apparent water age in Sanya Bay was estimated to be 0-13.2 days, with an average of

7.2±3.2 days. Based on the mass balance of $^{226}$Ra and $^{228}$Ra, SGD was calculated to be 2.76-5.03×10$^6$ m$^3$ d$^{-1}$ (or 4.3-7.7 cm d$^{-1}$), which accounted for more than half of the respective radium source flux into Sanya Bay. SGD associated dissolved inorganic nutrient fluxes into Sanya Bay were estimated to be 3.91-7.11×10$^5$ mol NO$_x$ d$^{-1}$, 5.03-

9.15×10$^5$ mol P d$^{-1}$, and 6.55-11.9×10$^5$ mol Si d$^{-1}$. The estuarine nutrients flux from the Sanya River was equivalent to the phosphate flux via SGD, but a few times smaller the nitrogen and silicate fluxes carried by SGD. SGD was also more important than atmospheric deposition and nitrogen fixation in the nutrients budget. Our results demonstrate that SGD contributed at least 38% phosphate, 90% nitrogen, and 83%

silicate in Sanya Bay. SGD could thus supply almost all nitrogen and silicate required by phytoplankton growth in the bay.

Key words: submarine groundwater discharge; radium isotopes; residence time; nutrients; China, Hainan Island, Sanya Bay

**1. Introduction**

Coastal waters are prone to deterioration under a global context of climate change and changes in ocean and land-source forces, such as acidification and hypoxia induced by upwelling [*Booth et al.*, 2012; *Feely et al.*, 2008; *Glenn et al.*, 2004; *Grantham et al.*, 2004; *Peterson et al.*, 2013] and eutrophication and hypoxia caused by increasing terrestrial nutrient loadings from catchment areas [*Zhang et al.*, 2010]. Among these interacting forces submarine groundwater discharge (SGD) has been recognized as an important carrier of water often featured with high concentrations of nutrients, dissolved inorganic and organic carbon, and metals [*Cai et al.*, 2003; *Charette et al.*, 2001; *Liu at al.*, 2012; *Moore*, 2010; *Moosdorf et al.*, 2015; *Porubsky et al.*, 2014]. Thus, SGD is a key factor to quantify in evaluating material budgets of any coastal system.

Naturally occurring radioactive radium isotopes ($^{223}$Ra, $^{224}$Ra, $^{226}$Ra, and $^{228}$Ra) have been widely used to trace SGD because they are not chemically active in coastal waters and their activities in SGD are at least an order of magnitude greater than in the receiving coastal waters [*Burnett and Dulaiova*, 2003; *Dulaiova et al.*, 2008; *Liu et al.*, 2012; *Moore*, 2010; *Schwartz*, 2003]. Radium is regenerated from decay of particle-reactive thorium isotopes and released from particles when encountering brackish or saline waters. The short-lived radium isotopes, $^{223}$Ra (half-life =11.4 days) and $^{224}$Ra (half-life =3.66 days), also work well in estimating apparent water ages on the shelf on time scales of a few to tens of days [*Gu et al.*, 2012; *Moore*, 2000; *Moore and Krest*, 2004].

Sanya Bay is a tropical bay located at the southern tip of Hainan Island, China in the northern South China Sea under the influence of the Southeast Asian monsoon (Fig. 1). Coral reefs account for 30% of its coastline [*Huang et al.*, 2003]. The Sanya

River flows into the bay in the northeast. Seasonal investigations in the bay demonstrate that the inner bay is influenced by the discharge of the Sanya River with relatively high nutrient levels, and the central and outer bay is dominated by oceanic forces from the South China Sea [*Wu et al.*, 2012a]. Our time-series studies demonstrate that tidally-driven SGD occurred at the Luhuitou fringing reef in the bay in a dry season, which caused coastal acidification and affected nutrient dynamics of the reef system [*Wang et al.*, 2014; 2017]. The flux of SGD into Sanya Bay based on mapping data, however, has never been reported.

[Figure]

Figure 1. Study area and sampling stations in Sanya Bay and the Sanya River estuary.

HK represents Hong Kong.

To quantify SGD and evaluate its geochemical impacts on Sanya Bay, a study was designed and implemented in Feb. 2012, using radium isotopes as SGD tracers. This study includes time-series observations at the Luhuitou fringing coral reef and a mapping investigation in the bay. The time-series observations were reported in *Wang*

*et al*. [2014; 2017]. The present work is focused on interpretations of the mapping data. Briefly, the residence time and the flux of SGD in Sanya Bay were estimated based on distributions of radium isotopes in the bay. Nutrients fluxes into the bay via

SGD were subsequently quantified and compared with other sources and sinks.

**2. Materials and Methods**

2.1. Study area

Sanya Bay has an average water depth of 16 m [*Huang et al.*, 2003] and irregular diurnal tides with a mean tidal range of 0.9 m [*Zhang*, 2001]. The annual mean surface water temperature is 26.8°C and the annual precipitation is around 1600-1800

mm [*Zhang*, 2001]. The Sanya River discharges into Sanya Bay in the northeast with an annual average discharge of 5.86 $m^3$ $s^{-1}$ [*Wang et al.*, 2005]. 95% of the rainfall occurs in May to October [*Li et al.*, 2013], so that the river discharge in the dry months is even lower than the annual average. Fringing reefs develop along the east coast and around the islands in the bay. Sanya Bay is oligotrophic under the influence of the northern South China Sea [*Wu et al.*, 2012b]. Multiple habitats, coral reefs, mangroves, mudflats, and rocky and sandy beaches, are present in the bay [*Huang et*

*al.*, 2003]. Holocene deposits of coral debris, sand, and silt surround the coast [*Zhao*

*et al.*, 1979]. The sediments in the bay are mostly sands (>60%) [*Che et al.*, 2010], composing a highly permeable surface quifer.

2.2. Sampling and measurements

Surface water samples were collected for radium using a plastic barrel in Sanya

Bay during Feb. 2-3, 2012 and at the lower Sanya River estuary station H1 on Feb. 4,

2012 (Fig. 1). Samples for nutrients were collected using a 5 L Niskin bottle at the same time in the Sanya River estuary in order to evaluate the estuarine export nutrients fluxes. Temperature and salinity were measured using a multiparameter sonde YSI 6600. The salinity was measured using the Practical Salinity Scale.

Groundwater samples were taken at domestic wells using a submersible pump.

Groundwater Station GW1 is about 50 m away from the coast. Details of GW1 are provided in *Wang et al* [2014] and samples were taken at this station every 2 hours from the morning of Feb. 7 to the morning of Feb. 8, 2012 for 24 hours to catch the diurnal variation of the groundwater. Station GW2 is about 100 m away from the coast and was sampled on Feb. 9, 2012. At this station the well was about 40 cm in diameter and 2.33 m deep and the water was 0.83 m deep. Samples for dissolved nitrate and nitrite, phosphate, silicate, and radium isotopes were taken at both groundwater stations.

Radium samples were passed through a 1 $\mu$m cartridge filter followed by a $MnO_2$- impregnated acrylic fiber (Mn-fiber) column to extract the dissolved radium [*Rama*

*and Moore*, 1996]. The Mn-fiber was measured for $^{223}$Ra and $^{224}$Ra with a radium delayed coincidence counter [*Moore and Arnold*, 1996] with an error less than 13%.

After the measurements were finished in two months, the Mn-fibers were leached for

$^{226}$Ra and $^{228}$Ra, which were then co-precipitated with $BaSO_4$ and measured in a germanium gamma detector (GCW4022, Canberra)[*Moore*, 1984] with an error less than 7%. To estimate radium desorbed from particles of the estuary water, total suspended matter (TSM) was collected at Station P1 on pre-weighed and pre- combusted 47-mm-diameter GF/F filters (pore size of 0.7 μm) and measured by weighing after drying.

Nutrient samples were filtered through 0.45 μm cellulose acetate membranes and preserved with 1-2‰ chloroform. One filtrate was stored at 4°C before measurement for silicate, and one was kept at -20°C for nitrate, nitrite, and phosphate measurements. In the laboratory, nitrate, nitrite, silicate and phosphate were measured with a Technicon AA3 Auto-Analyzer (Bran-Luebbe, GmbH) following the same methods in *Han et al.* [2012]. The analytical precision was better than 1% for nitrate and nitrite, 2% for phosphate, and 2.8% for silicate.

2.3. Radium mass-balance model and residence time estimation

The decay of the long-lived radium isotopes, $^{226}$Ra (half-life = 1600 yrs) and $^{228}$Ra (half-life = 5.75 yrs), can be ignored in studying coastal and estuarine processes

[*Moore et al.*, 2006]. Under the assumption of steady state of the system investigated, long-lived radium loss via mixing was equal to gains from river, SGD, and sediment diffusion, i.e.,

$$F_R \cdot {}^iRa_R + {}^iF_{sed} \cdot A_B + F_R \cdot f_d \cdot {}^iRa_p \cdot C_{TSM} + F_{SGD} \cdot {}^iRa_{GW} = V_B \cdot ({}^iRa_B - {}^iRa_O) \cdot \frac{1}{\tau} \qquad (1)$$

where on the left-hand side are the source terms: the first term represents the dissolved radium flux from the river, where $F_R$ is the river water discharge, ${}^iRa_R$ is the activity of dissolved ${}^iRa$ of the estuary water, $i$=226 and 228; the second term represents the sediment diffusion flux of radium, where ${}^iF_{sed}$ is the areal diffusive flux of ${}^iRa$ from the sediments, and $A_B$ is the sediment surface area of the bay investigated; the third term represents the desorbed radium flux from the river, where $f_d$ is the fraction of radium exchangeable from particles, $^iRa_p$ is the activity of $^iRa$ on particles, and $C_{TSM}$ is the concentration of TSM of the estuary water; and the fourth term represents the radium flux via SGD, where $F_{SGD}$ is the SGD flux, and $^iRa_{GW}$ is the average activity of dissolved $^iRa$ of the groundwater; on the right-hand side are the sink terms: where $V_B$ is the volume of the bay under investigation, $^iRa_B$ is the average activity of dissolved $^iRa$ in the bay, $^iRa_O$ is the activity of dissolved $^iRa$ of the ocean water, and $\tau$ is the residence time in the bay.

The residence time in the bay can be estimated by the activity ratio of $^{224}$Ra and

$^{228}$Ra under the assumption of steady state as derived by *Moore et al.* [2006]:

$$\tau = \frac{F\left(\dfrac{^{224}Ra}{^{228}Ra}\right) - I\left(\dfrac{^{224}Ra}{^{228}Ra}\right)}{I\left(\dfrac{^{224}Ra}{^{228}Ra}\right) \cdot \lambda_{224}} \qquad (2)$$

where $F\left(\dfrac{^{224}Ra}{^{228}Ra}\right)$ is the ratio of the flux of $^{224}$Ra over that of $^{228}$Ra into the system, equivalent to the activity ratio of $^{224}$Ra to $^{228}$Ra of the flux into the system, and

$I\left(\dfrac{^{224}Ra}{^{228}Ra}\right)$ is the ratio of the inventory of $^{224}$Ra over that of $^{228}$Ra in the system, which is equal to the activity ratio of $^{224}$Ra to $^{228}$Ra in the system.

**3. Results**

3.1. Radium isotopes in Sanya Bay

Activities of $^{223}$Ra ranged 0.4-1.8 dpm 100 L$^{-1}$ (i.e., 0.07-0.3 Bq m$^{-3}$), decreasing offshore and southward with the maximum in the north of the bay (Fig. 2a). The activity of $^{228}$Ra showed a similar pattern, varying in the range 23.1-38.0 dpm 100 L$^{-1}$

(Fig. 2d). $^{224}$Ra and $^{226}$Ra demonstrated the highest activities in the northeast bay off the Sanya River estuary. The range of activity was 11.9-42.6 dpm 100 L$^{-1}$ for $^{224}$Ra and 9.6-11.9 dpm 100 L$^{-1}$ for $^{226}$Ra (Figs. 2b,c). In general, activities of radium isotopes were higher in the northern Sanya Bay and outside the Sanya River estuary, coincident with lower salinities of 33.60-33.62 at these stations (Table 1). These higher radium signals were reflective of the Sanya River plume and other land sources.

| Station | Latitude | Longitude | Water Depth (m) | Temp (°C) | Salinity | $^{223}$Ra | σ | $^{224}$Ra | σ | $^{226}$Ra | σ | $^{228}$Ra | σ |
|---|---|---|---|---|---|---|---|---|---|---|---|---|---|
| | | | | | | dpm 100 L$^{-1}$ | | | | | | | |
| J1 | 18.2718 | 109.4565 | 8 | 22.80 | 33.60 | 1.79 | 0.20 | 33.86 | 0.36 | 10.81 | 0.45 | 33.73 | 1.30 |
| J2 | 18.2623 | 109.4423 | 9 | 22.66 | 33.62 | 0.80 | 0.13 | 21.52 | 0.68 | 10.50 | 0.39 | 31.74 | 1.09 |
| J3 | 18.2531 | 109.4298 | 12 | 22.70 | 33.64 | 1.56 | 0.16 | 30.04 | 0.38 | 10.91 | 0.50 | 37.95 | 1.44 |
| J4 | 18.2409 | 109.4118 | 11 | 22.81 | 33.70 | 0.60 | 0.14 | 11.92 | 0.40 | 10.78 | 0.39 | 28.24 | 1.02 |
| J5 | 18.2261 | 109.3909 | 15 | 22.90 | 33.70 | 0.52 | 0.15 | 17.61 | 0.89 | 10.19 | 0.41 | 28.02 | 1.10 |
| W1 | 18.2555 | 109.4832 | 5 | 23.12 | 33.70 | 1.36 | 0.23 | 26.66 | 0.39 | 12.04 | 0.49 | 30.35 | 1.23 |
| W2 | 18.2466 | 109.4672 | 12 | 22.93 | 33.72 | 0.83 | 0.13 | 12.88 | 0.32 | 10.92 | 0.41 | 26.22 | 1.02 |
| W3 | 18.2306 | 109.4413 | 16 | 22.97 | 33.89 | 0.59 | 0.13 | 11.93 | 0.53 | 10.63 | 0.43 | 23.11 | 1.06 |
| W4 | 18.2154 | 109.4244 | 5 | 23.12 | 33.70 | 0.52 | 0.11 | 12.82 | 0.30 | 10.73 | 0.39 | 24.03 | 0.98 |
| P1 | 18.2355 | 109.4940 | 5 | 22.98 | 33.62 | 1.22 | 0.19 | 42.60 | 0.85 | 11.93 | 0.54 | 28.87 | 1.34 |
| P2 | 18.2296 | 109.4797 | 11 | 23.01 | 33.67 | 0.97 | 0.17 | 23.57 | 0.34 | 10.56 | 0.49 | 31.94 | 1.27 |
| P3 | 18.2213 | 109.4660 | 16 | 22.75 | 33.84 | 0.54 | 0.13 | 15.49 | 0.59 | 10.99 | 0.41 | 24.26 | 1.08 |
| P4 | 18.2105 | 109.4464 | 12 | 22.71 | 33.89 | 0.77 | 0.11 | 15.32 | 0.36 | 10.83 | 0.32 | 28.14 | 0.85 |
| P5 | 18.1931 | 109.4296 | 26 | 22.69 | 33.81 | 1.38 | 0.11 | 18.51 | 0.65 | 10.95 | 0.33 | 28.90 | 0.89 |
| L1 | 18.2219 | 109.4812 | 7 | 22.76 | 33.84 | 0.65 | 0.10 | 13.65 | 0.40 | 9.71 | 0.42 | 26.42 | 1.15 |
| L2 | 18.2193 | 109.4812 | 11 | 22.81 | 33.85 | 0.74 | 0.11 | 22.99 | 0.67 | 11.98 | 0.44 | 26.33 | 1.16 |
| L3 | 18.2201 | 109.4749 | 12 | 22.79 | 33.84 | 0.78 | 0.11 | 14.49 | 0.28 | 9.77 | 0.41 | 25.24 | 1.06 |
| L4 | 18.2105 | 109.4674 | 20 | 22.76 | 33.85 | 0.70 | 0.11 | 13.82 | 0.42 | 10.59 | 0.42 | 25.53 | 1.13 |
| L5 | 18.2111 | 109.4582 | 21 | 22.74 | 33.85 | 1.17 | 0.13 | 17.63 | 0.33 | 11.27 | 0.38 | 25.90 | 0.96 |
| L6 | 18.1965 | 109.4694 | 23 | 22.79 | 33.82 | 0.84 | 0.12 | 16.34 | 0.38 | 10.21 | 0.43 | 23.77 | 0.95 |
| L7 | 18.1966 | 109.4601 | 32 | 22.83 | 33.86 | 0.84 | 0.18 | 14.26 | 0.76 | 10.53 | 0.41 | 26.03 | 1.08 |
| L8 | 18.1964 | 109.4476 | 25 | 22.78 | 33.88 | 0.43 | 0.15 | 12.18 | 0.40 | 9.58 | 0.43 | 27.01 | 1.20 |
| H1 | 18.2348 | 109.4977 | nd* | 22.88 | 31.70 | 1.69 | 0.50 | 64.75 | 0.74 | 15.45 | 0.70 | 43.75 | 1.85 |

*nd– not determined

Table 1. Sampling stations and data for surface water in Sanya Bay and the lower

Sanya River estuary in Feb. 2012.

[Figure]

Figure 2. Surface distributions of radium isotopes (in dpm 100 L$^{-1}$) in Sanya Bay, (a) $^{223}$Ra, (b) $^{224}$Ra, (c) $^{226}$Ra, and (d) $^{228}$Ra.

**3.2. Parameters of the estuary water and of the groundwater**

The salinity in the investigated Sanya River estuary increased from 6.06 downstream to 31.70 at the estuary outlet. Temperature ranged from 23.12-24.00. Nutrients decreased consistently with salinity for oxidized inorganic nitrogen (NO$_x$) and silicate, from 36.6 to 6.72 μM for NO$_x$ with nitrite accounting for one third of NO$_x$ and from 271 to 30.1 μM for silicate (Fig. 3a). For phosphate a general decreasing trend was present (Fig. 3b), however, the peak concentration, 11.0 μM, appeared at the mid-salinity station H8, where the salinity was 15.60, and the minimum concentration, 1.45 μM, showed at Station H3, where the salinity was 28.91. The deviation from conservative mixing of phosphate in the mid-salinity in estuaries has been proposed to be due to particle sorption/desorption [*Froelich et al.,*

1982; *van der Zee et al.*, 2007]. The estuarine station H1 had a salinity of 31.70 and relatively high activities of radium isotopes (in dpm 100 L$^{-1}$) compared with the bay water, 1.7 for $^{223}$Ra, 64.8 for $^{224}$Ra, 15.5 for $^{226}$Ra, and 43.8 for $^{228}$Ra. TSM of the estuary water was 25.3 mg L$^{-1}$.

[Figure]

Figure 3. Concentrations of nutrients against salinity in the Sanya River estuary, (a)

oxidized inorganic nitrogen (NO$_x$) and silicate (b) phosphate.

A weekly observation of temperature and salinity at groundwater Station GW1

indicated that groundwater properties were relatively constant with time, without apparent tidal resonances and the salinity varied in the range of 20.06-20.49 [*Wang et*

*al.*, 2014]. NO$_x$ was mostly nitrate with nitrite less than 0.1% (i.e., <0.1 μM). The average concentrations of NO$_x$, phosphate, and silicate (in μM) were 141.5±14.2,

1.68±0.53, and 237.2±2.2, with n=13, respectively. The average activities of radium isotopes (in dpm 100 L$^{-1}$) were 30.6±7.2 for $^{223}$Ra, 624.2±25.8 for $^{224}$Ra, 245.9±25.9

for $^{226}$Ra, and 434.9±17.3 for $^{228}$Ra. At Station GW2 the salinity was 0.20. The activities of radium isotopes (in dpm 100 L$^{-1}$) were much lower than at Station GW1,

1.96 for $^{223}$Ra, 62.4 for $^{224}$Ra, 17.7 for $^{226}$Ra, and 42.9 for $^{228}$Ra; while concentrations of nutrients were about twice higher for NO$_x$ than at Station GW1, twice as high for silicate, but half as much for phosphate.

**4. Discussion**

4.1. Residence time in Sanya Bay

[Figure]

Figure 4. Activities of $^{224}$Ra vs. $^{223}$Ra (a) and $^{228}$Ra vs. $^{226}$Ra (b) of Sanya Bay water, the lower Sanya River estuary water, and nearby groundwater in Feb. 2012.

Dissolved radium in Sanya Bay appeared to have the same source as that of the estuary water and of the groundwater with $^{224}$Ra vs. $^{223}$Ra and $^{228}$Ra vs. $^{226}$Ra falling not far from a linear line (Fig. 4). The activity ratio of $^{224}$Ra/$^{228}$Ra ranged 0.42-1.48 in

Sanya Bay with the maximum occurring at Station P1 outside the Sanya River estuary, with higher values in the north and northeast of the bay (Fig. 5a), indicating sources of radium from the coastline. The intrusion of the northern South China Sea water into the bay caused the lower activity ratio of $^{224}$Ra/$^{228}$Ra in the south of the bay. In terms of the sources of radium into Sanya Bay, the activity ratio of $^{224}$Ra/$^{228}$Ra was almost the same for the Sanya River plume and SGD, 1.48 for Sanya River estuary water and 1.44±0.07 for the groundwater. Considering that the radium flux from sediment diffusion is usually less than SGD [*Liu et al.*, 2012; *Moore et al.*,

2006], the residence time in the bay was estimated using Eq. (2), taking 1.48 to represent the activity ratio of radium input fluxes from the river plume and SGD. The residence time ranged 0-13.2 days in Sanya Bay with an average of 7.2±3.4 days, relatively short near the north and northeast coast of the bay and increasing offshore (Fig. 5b).

[Figure]

Figure 5. Activity ratio of $^{224}$Ra/$^{228}$Ra (a) and residence time ($\tau$) (b) in Sanya Bay in

Feb. 2012.

4.2. SGD estimation using radium isotopes

To estimate SGD into Sanya Bay, the mass balance of $^{226}$Ra and $^{228}$Ra was set up as illustrated in Eq. (1). The average salinity of the bay water was 33.77±0.10. Thus, groundwater Station GW1, a well much closer to the coast, where the average salinity was 20.22, was more representative of SGD water directly interacting with the bay water. Therefore, data from Station GW1 were taken as the SGD end-member. The annual Sanya River discharge was taken into Eq. (1) for a minimum SGD estimate.

The parameters at the lower estuarine station H1 were taken to calculate the river/estuarine contribution of radium to the bay. Diffusive fluxes of radium were taken from the literature. Radium data at an offshore station (110 ºE, 18 ºN) were taken to represent the ocean water radium. All the parameters used in Eq. (1) to estimate SGD are listed in Table 2 and sources and sinks of radium in the bay were quantified and are listed in Table 3. The SGD flux was estimated to be $2.67 \times 10^6$ m$^3$ d$^-$

$^1$ (or 4.1 cm d$^{-1}$) based on $^{226}$Ra and $5.01 \times 10^6$ m$^3$ d$^{-1}$ (or 7.7 cm d$^{-1}$) based on $^{228}$Ra, which accounted for 98% of the respective radium source flux into Sanya Bay. This was comparable to the SGD rate along the eastern coast of Hainan Island and in other embayments (Table 4). The rate estimated using mapping data of long-lived radium isotopes in the bay fell in the range of seepage rates derived from time-series observations of $^{226}$Ra in a coastal station in Sanya Bay, 0-44 cm d$^{-1}$ [*Wang et al.*,

2014].

| | Parameter | | Value | Unit | Reference |
|---|---|---|---|---|---|
| | $F_R$ | River discharge | 5.86 | m$^3$ s$^{-1}$ | *Wang et al.*, 2005 |
| Estuary | $^{226}$Ra$_R$ | Estuary water $^{226}$Ra | 15.45 | dpm 100 L$^{-1}$ | This study |
| | $^{228}$Ra$_R$ | Estuary water $^{228}$Ra | 43.75 | | |
| | $C_{TSM}$ | Concentration of total | 25.33 | mg l$^{-1}$ | |

| | | suspended matter | | | |
|---|---|---|---|---|---|
| | $f_d$ | Fraction of desorbed radium from particles | 0.43 | ----- | *Wang et al.,* 2015 |
| | $^{226}Ra_p$ | $^{226}Ra$ on particles | 2.5 | dpm g$^{-1}$ | *Krest and Moore,* 1999 |
| | $^{228}Ra_p$ | $^{228}Ra$ on particles | 2.09 | | |
| Sediment | $^{228}F_{sed}$ | $^{228}Ra$ diffusive flux | 2.1 | dpm m$^{-2}$ d$^{-1}$ | *Charette et al.,* 2001 |
| | $^{226}F_{sed}$ | $^{226}Ra$ diffusive flux | 0.27 | | |
| Groundwater | $^{226}Ra_{GW}$ | Groundwater $^{226}Ra$ | 255.8 | dpm 100 L$^{-1}$ | This study |
| | $^{228}Ra_{GW}$ | Groundwater $^{228}Ra$ | 454.0 | | |
| Sanya Bay | $^{226}Ra_B$ | Bay water $^{226}Ra$ | 10.75 | | |
| | $^{228}Ra_B$ | Bay water $^{228}Ra$ | 27.81 | | |
| | $V_B$ | Volume of the bay investigated | $1.04\times10^9$ | m$^3$ | |
| | $A_B$ | Surface area of the bay investigated | $6.49\times10^7$ | m$^2$ | |
| | $\tau$ | Residence time | 7.24 | day | |
| Ocean | $^{226}Ra_O$ | Ocean water $^{226}Ra$ | 5.92 | dpm 100 L$^{-1}$ | |
| | $^{228}Ra_O$ | Ocean water $^{228}Ra$ | 11.70 | | |

Table 2. Parameters used in the mass balance Eq. (1) of $^{226}Ra$ and $^{228}Ra$.

| Radium | | | Formula in Eq.(1) | Value | Unit |
|---|---|---|---|---|---|
| $^{226}Ra$ | Sources | Sanya River | $F_R\cdot{}^{226}Ra_R$ | $7.82\times10^7$ | dpm d$^{-1}$ |
| | | | $F_R\cdot f_d\cdot{}^{226}Ra_p\cdot C_{TSM}$ | $1.31\times10^7$ | |
| | | Sediment diffusion | $A_B\cdot{}^{226}F_{sed}$ | $1.75\times10^7$ | |
| | | Groundwater | $F_{SGD}\cdot{}^{226}Ra_{GW}$ | $6.82\times10^9$ | |
| | Sink | Mixing | $V_B\cdot({}^{226}Ra_B-{}^{226}Ra_O)/\tau$ | $6.93\times10^9$ | |
| $^{228}Ra$ | Sources | Sanya River | $F_R\cdot{}^{228}Ra_R$ | $2.22\times10^8$ | |
| | | | $F_R\cdot f_d\cdot{}^{228}Ra_p\cdot C_{TSM}$ | $1.10\times10^7$ | |
| | | Sediment diffusion | $A_B\cdot{}^{228}F_{sed}$ | $1.36\times10^8$ | |
| | | Groundwater | $F_{SGD}\cdot{}^{228}Ra_{GW}$ | $2.27\times10^{10}$ | |
| | Sink | Mixing | $V_B\cdot({}^{228}Ra_B-{}^{228}Ra_O)/\tau$ | $2.31\times10^{10}$ | |

Table 3. Sources and sinks of long-lived radium ($^{226}Ra$ and $^{228}Ra$) in Sanya Bay.

| Region | SGD rate (cm d$^{-1}$) | References |
|---|---|---|
| Manila Bay, Philippines | 0-26 | *Taniguchi et al.,* 2008 |
| Jamaica Bay, USA | 1.5-17 | *Beck et al.,* 2007 |

| Masan Bay, Korea | 6.1-7.1 | *Lee et al.*, 2009 |
|---|---|---|
| Yeogil Bay, Korea | 20 | *Kim et al.*, 2007 |
| Eastern coast of Hainan Island, China | 10-29 | *Ji et al.*, 2013 |
| Sanya Bay, China | 4.1-7.7 | This study |

Table 4. The SGD flux in Sanya Bay compared with SGD rates in other embayments and along the eastern Hainan Island.

4.3. Nutrients fluxes via SGD into Sanya Bay and their contributions to the nutrients budgets

Nutrients fluxes via SGD into Sanya Bay were calculated using the flux of SGD

estimated from surface distributions of long-lived radium in the bay multiplied by nutrients concentrations at the groundwater Station GW1. Thus, nutrients fluxes via

SGD were $4.48\text{-}8.42 \times 10^3$ mol d$^{-1}$ for phosphate, $3.77\text{-}7.09 \times 10^5$ mol d$^{-1}$ for NO$_x$, and

$6.32\text{-}11.9 \times 10^5$ mol d$^{-1}$ for silicate. Sanya Bay is relatively oligotrophic with concentrations of nutrients in the range of below the detection limit (BDL) to 0.17 µM

for phosphate, BDL to 1.13 µM for NO$_x$, and 4.06-7.92 µM for silicate [*Wang et al.*,

2017]. The inventory of nutrients in Sanya Bay was estimated, taking the average concentration of nutrients in the bay and multiplied by the water volume under investigation, to be $4.58 \times 10^4$ mol P, $3.83 \times 10^5$ mol NO$_x$, and $5.40 \times 10^6$ mol Si. The inventory was then divided by the residence time in the bay, 7.24 d, and a removal rate of nutrients by mixing was estimated to be $6.33 \times 10^3$ mol P d$^{-1}$, $5.29 \times 10^4$ mol NO$_x$

d$^{-1}$, and $7.46 \times 10^5$ mol Si d$^{-1}$. Comparisons with SGD-associated nutrients fluxes indicated that SGD could supply all NO$_x$ and almost all phosphate and silicate removed by mixing in Sanya Bay. The average planktonic primary production in

Sanya Bay in winter is 39.36 mmol C $m^{-2}$ $d^{-1}$ [*Dong et al.*, 2008]. Assuming an uptake ratio of C:N:P:Si of 106:16:1:15 [*Brzezinski*, 1985; *Redfield*, 1960], the corresponding nutrient uptake rates would be $2.41 \times 10^4$ mol $d^{-1}$ for P, $3.86 \times 10^5$ mol $d^{-1}$ for N, and

$3.61 \times 10^5$ mol $d^{-1}$ for Si. SGD seemed to provide more than enough N and Si and at least 19% of the P necessary to support this planktonic primary production. In addition, nitrite, nitrate, and phosphate at offshore stations were below detection limits [*Wang et al.*, 2017], indicating that the ocean provided negligible, if any, nutrients to Sanya Bay. The average nitrogen fixation rate in the bay is 0.14 mmol $m^{-2}$

$d^{-1}$ in winter [*Dong et al.*, 2008], at most 2% equivalent to that contributed by SGD.

The estuarine export nutrients fluxes from the Sanya River estuary were estimated, using an effective concentration multiplied by the annually-average river discharge, to be $7.33 \times 10^3$ mol $d^{-1}$ for phosphate, $2.52 \times 10^4$ mol $d^{-1}$ for $NO_x$, and $1.28 \times 10^5$ mol $d^{-1}$

for silicate. The effective concentration was the *y* intercept of a linear regression of the concentration in the estuary against salinity at mid to high salinity [*Officer*, 1979].

As shown in Fig. 3, the linear regressions were significant for these nutrients with

$R^2 > 0.9$ and the effective concentration was 16.0 $\mu M$ for phosphate, 54.9 $\mu M$ for $NO_x$, and 277 $\mu M$ for silicate. The estuarine export phosphate flux was comparable to the

SGD-associated flux, while the fluxes of $NO_x$ and phosphate from the Sanya River estuary were less than that contributed by SGD. Another source of nutrients is atmospheric deposition. Since there was no rain during the two weeks before our sampling, a higher dry deposition rate of nitrogen for the south China from the literature, $9.72 \times 10^{-5}$ mol N $m^{-2}$ $d^{-1}$ [*Wai et al.*, 2010], was considered, which gave a deposition flux of $6.31\times10^3$ mol N $d^{-1}$. The deposition flux is about two orders of magnitude smaller than SGD-contributed nitrogen. Thus, SGD is a main nutrient contributor to Sanya Bay at least as important as the Sanya River.

In the nutrients budgets of Sanya Bay (Fig. 6), the source terms include the Sanya

River estuarine export, SGD, atmospheric deposition, and nitrogen fixation. The sink terms are ocean mixing and biological uptake. The total sink is $4.39\times10^5$ mol N $d^{-1}$,

$2.78\times10^4$ mol P $d^{-1}$, and $1.11\times10^6$ mol Si $d^{-1}$, while the total source is $4.18\text{-}7.50\times10^5$

mol N $d^{-1}$, $1.18\text{-}1.58\times10^4$ mol P $d^{-1}$, and $0.76\text{-}1.32\times10^6$ mol Si $d^{-1}$. Apparently, the source and sink terms of nitrogen and silicate can be balanced in Sanya Bay. A deficit in phosphate is present. At least $1.20\times10^4$ mol P $d^{-1}$ is required to fill the gap. We propose two reasons for this deficit: a) benthic flora of about 150 species were found in Sanya Bay [*Titlyanov et al.*, 2015] and macroalgae usually demonstrate an N:P

ratio of about 30 in their tissues [*Atkinson and Smith*, 1983]; if this ratio were considered in estimating the biological uptake rate of phosphate based on the nitrogen uptake rate, a much lower biological uptake rate of phosphate would have been obtained; and b) benthic release of phosphorus due to remineralization or grazing of organic matter may be a phosphate source.

Nutrients carried by SGD contributed 90-95% nitrogen, 38-53% phosphate, and

83-90% silicate to the nutrients source of Sanya Bay. Our results substantiate the regulation of SGD on nutrient composition in a coral reef system of Sanya Bay found in our time-series studies [*Wang et al.*, 2017]. Nutrient enrichments have caused worldwide coastal environmental issues of eutrophication and hypoxia [*Chislock and*

*Doster*, 2013; *Howarth et al*., 2011]. As a major nutrient source, with frequencies and areas of eutrophication and associated hypoxia increasing around the world coast

[*Diaz and Rosenberg*, 2008], SGD and its associated material fluxes need to be monitored in the long term in environmental protection programs of any coastal ecosystems.

[Figure]

Figure 6. Nutrients budgets in Sanya Bay. Unit is in mol d$^{-1}$.

**5. Conclusions**

Contribution of SGD-associated nutrients to the nutrients budget in Sanya Bay in the dry season was investigated for the first time using naturally occurring radium isotopes as SGD tracers. The following was concluded from this study:

a)    In Sanya Bay, radium isotopes ($^{223}$Ra, $^{224}$Ra, $^{226}$Ra, and $^{228}$Ra) had higher

  activities along the north coast and in the northeast near the Sanya River

  estuary, indicating sources of radium from the coast and the river.

b)    The residence time in Sanya Bay ranged 0-13.2 days, with an average of

  7.2±3.4 days, relatively short near the north and northeast coast of the bay

  and increasing offshore.

c)    SGD associated dissolved inorganic nutrient fluxes into Sanya Bay were

  estimated to be 3.91-7.11×10$^5$ mol NO$_x$ d$^{-1}$, 5.03-9.15×10$^5$ mol P d$^{-1}$, and

$6.55\text{-}11.9\times10^5$ mol Si $d^{-1}$. SGD could satisfy all nitrogen and silicate requirements and 20% of phosphate requirement by phytoplankton growth in Sanya Bay. The nutrients fluxes via SGD are at least comparable to the estuarine export fluxes from the Sanya River.

d)   SGD is a major source of nitrogen and silicate and contributes at least 38%

phosphate in Sanya Bay.

Acknowledgements

We thank the crew on the ship QiongLinGao 02706 and Junde Dong for arranging local logistic support at the Tropical Marine Biological Research Station in Hainan,

Chinese Academy of Sciences. Wenping Jing, Zhouling Zhang, and Yi Xu helped in sample collection. Daochen Zhao measured the long-lived radium isotopes. This work was supported by MOST (2015CB954001) and the National Natural Science

Foundation of China (41576074).

---

## Referee Report (RR1)

**Review of the manuscript entitled "Tidal variability of nutrients in a coastal coral reef system influenced by groundwater" by Wang et al. (BG 2017 - 156)**

Nitrification has been considered to be a two-step process consisting of ammonia oxidation (ammonium to nitrite) followed by nitrite oxidation (nitrite to nitrate), and each step is carried out by different microbes. Not much data are available on the rates of nutrient uptake at the low concentrations of nutrients occurring naturally in reef waters. Predictions of sustainable uptake/release rates in the natural environment based on uptake from enriched seawater with submarine groundwater discharge are questionable without detailed measurements and models. In this manuscript measurement of nitrite, nitrate without ammonia are reported which will not provide a complete picture in the absence of microbial processes that controls the DIN make up in estuarine/coral system. Corals may be capable of adaptive changes in uptake kinetics dependent on nutrient availability. However, the rate of nitrogen acquisition appears to be influenced on a diel cycle, presumably due to depletion of photosynthetic products during the night. Towards this, the manuscript is not strong in presenting the results and conclusions. It is, however, necessary to acknowledge the controversy and to address the potential biases associated with the choices made in the calculations.

Almost 80% of SGD, calculated using the salinity difference between time-series station CT and the close seawater station, contributes very little increase nutrient concentrations and attributed due to biological processes as mentioned in the text. This contradicts the sentence on Page 3 line 25. What is the concentration of silicate, phosphate and nitrogen species in the river water and groundwater? Including this information will improve the reliability of the data discussed.

I believe this manuscript is an important contribution to the field, however, much editing is needed before publication.

The specific comments are listed below:

Page 4 line 11: Prove volume of seawater

Page 4 line 19: Too simplistic. Delete this sentence.

Page 5 line 2:  Delete "using Microsoft Excel (2007)"

Page 5 line 28: Check the values and correct the sentence.

Page 6 line 23: spelling! Write 'vertical' instead of 'verticle'

Page 8 line 3: Therefore we propose…. Not clear. Rewrite the sentence.

Page 8 line 15: During the flood tide…. This sentence is not clear. Rewrite it.

Page 8 line 20: The slope and constant values quoted in the equations 2 to 3 of section 4.2 are with more than significant digits of the salinity and nutrient concentrations used for getting them. Also you should include error on these values to show whether they are significantly different beyond the precision of measurement of the nutrients. Also correct them in Fig. 8. Delta P bio values are especially not different from the precision of measurement.

---

## Author Response (AR2)

**Response to the comments on "Tidal variability of nutrients in a coastal coral reef system influenced by groundwater"**

Responses are in blue with page and line numbers provided where changes are made in the revision.

Submitted on 24 Oct 2017
Anonymous Referee #2

**Anonymous during peer-review: Yes** No

**Anonymous in acknowledgements of published article: Yes** No

**Recommendation to the editor**

| | |
|---|---|
| **1) Scientific significance**
Does the manuscript represent a substantial contribution to scientific progress within the scope of this journal (substantial new concepts, ideas, methods, or data)? | Excellent **Good** Fair Poor |
| **2) Scientific quality**
Are the scientific approach and applied methods valid? Are the results discussed in an appropriate and balanced way (consideration of related work, including appropriate references)? | Excellent Good **Fair** Poor |
| **3) Presentation quality**
Are the scientific results and conclusions presented in a clear, concise, and well structured way (number and quality of figures/tables, appropriate use of English language)? | Excellent Good **Fair** Poor |

For final publication, the manuscript should be

**accepted as is**

accepted subject to **technical corrections**

accepted subject to **minor revisions**

**reconsidered after major revisions**

> **I would like to review the revised paper.**
>
> I am **not** willing to review the revised paper.

**rejected**

**Suggestions for revision or reasons for rejection (will be published if the paper is accepted for final publication)**

Comments:

The authors conclude two key points: nutrient concentrations at station CT reflected the

mixing of nitrite-deplete, nitrate and phosphate-rich less saline groundwater and nitrite-rich, nitrate and phosphate-deplete saline offshore seawater and quantified variations in oxidized nitrogen and phosphate contributed by biological processes. This manuscript addresses the small scale effects of groundwater discharge on nutrient loads in Sanya Bay and should be very important on understanding nutrient variability. However, I am not sure whether because I don't read the other published and submitted papers closely related to this manuscript, I still have some questions for this revision.

In the authors' response, they mentioned that "at Station CT, which is about 30 m away from the coast with water depth of 0.7-2.1 m, the groundwater discharge is from both the seabed and the coast". If the groundwater discharge is non-point source, the groundwater discharge can be recognized in the bay or just at station CT? In addition, water masses mixing besides groundwater discharge exists in the bay?

Response: the groundwater discharge can be recognized in the bay from radium distributions as the manuscript of the groundwater in the bay presents (the manuscript was provided earlier). The processes related with the groundwater discharge can be better recognized through time-series observations, though. In the bay besides groundwater there exists mixing of river water and seawater. However, the Sanya River plume is constrained to the northeast of the bay in winter as the salinity distribution indicates (Fig. 1, Page 6 Line 19-26). So at Station CT only mixing of groundwater and seawater exists.

Based on the relationship between nutrients and salinity, the authors tried to estimate the effects of biological processes. The most important is to confirm the two end members represent the groundwater discharge and offshore seawater.

Response: we agree with the reviewer. We have confirmed the two end members in the manuscript based on salinity distribution. In the revision we add salinity profiles to further confirm it (Fig. 1b, Page 6 Line 19-26).

Other comments:

P6, Discussion 4.1: The authors concluded that "the Sanya River plume affected the northeast of the bay with little impact on Station CT and the only source of freshwater at this site in February would be groundwater discharge", I cannot understand how this conclusion comes out and what is the evidence, and why the minimum salinity scale for station CT (Figure 3) is lower than Sanya bay (Figure 1). The latter should cover station CT.

Response: to clarify the source of freshwater at Station CT is groundwater, in the revision more explanations, including vertical salinity profiles, are provided (Fig. 1b, Page 6 Line 19-26). Both the horizontal and vertical distributions of salinity (Fig. 1) indicate that the Sanya River plume is constrained to the northeast of Sanya Bay and has no influence on Station CT.

At Station CT the salinity is in the range of 33.43-33.67, which is based on time-series observations from spring to neap tide. The salinity distribution in Sanya Bay is based on mapping of the bay, which is a snapshot. Because the snapshot didn't occur at the lowest tide of the spring tide, the salinity distribution does not cover the minimum salinity at Station CT.

P8, Discussion 4.2: The authors contribute the correlation relationship between nutrients and salinity to "mixing between the groundwater discharge and the offshore seawater". Why it is not mixing of different water masses? Thus, the two assumptions for estimate generation

and consumption of NOx and phosphate need evidence! Even if we take these assumptions, the calculated $\Delta NOxbio$ and $\Delta Pbio$ contributed by biological processes should have similar ratios during both spring and neap tides. However, the fact is that "the relationship between $\Delta NO3bio$ and $\Delta Pbio$ during the spring tide differed from that during the neap tide", which is unbelievable. Why primary producers assimilate NOx and phosphate with different ratios from the offshore seawater and groundwater discharge or the other sources. I guess the main problem is that both NO2 and NO3 concentrations were low and ammonium may be an important composition and influence the N/P ratio. But the authors don't consider the contribution of ammonium.

Response: in the revision more explanations, including vertical salinity profiles, are provided to confirm that the source of freshwater at Station CT is groundwater (Fig. 1b, Page 6 Line 19-26). The reviewer is right in that ammonium contributes dissolved inorganic nitrogen (DIN). Unfortunately, ammonium data are not available for this time-series observation. So we limit our discussion to $NO_x$. But we expect that the ammonium concentration must have varied during spring to neap tide as the other nutrients. The changes in the ratio of $\Delta NO_{xbio}$ and $\Delta P_{bio}$ from spring to neap tide may have reflected changes in ammonium. In the revision we add that $NO_x$ is not equivalent to DIN and point out the limitation of not including ammonium (Page 9 Line 20-23).

P7, Lines 2-6, I totally lost. The authors mentioned "The daily maximum concentration of NOx, phosphate, and silicate appeared in the day time at relatively low tides……. During the neap tide, ……The daily maximum concentration of NOx and phosphate appeared around the mid-night, when a flood tide appeared". Then the authors conclude that "This pattern reflected dominance of biological processes". What "this pattern" means? and why?

Response: "this pattern" means "the daily maximum of $NO_x$ and phosphate appeared around the mid-night, when a flood tide appeared". As mentioned in this paragraph, the groundwater discharge is characterized by higher nitrate and phosphate and lower nitrite than the offshore seawater. Also nitrate dominates $NO_x$. So groundwater discharge is characterized by higher $NO_x$ and phosphate. When biological processes control the system, the maximum concentrations of nutrients should appear at night when respiration predominates and the minimum appears during the day time when uptake of nutrients occurs. When groundwater plays a more important role in the system, groundwater discharge is greater at low tides than at high tide due to its tidal pumping feature so that $NO_x$ and phosphate should be higher at low tides and lower at high tides. In the revision this explanation is added (Page 7 Line 6-14).

P5, Please explain why nutrient variabilities were different, such as nitrite; nitrate and phosphate; silicate, which derived from different sources or others?

Response: as discussed in Discussion 4.1 nutrient variations are controlled by mixing of groundwater and offshore seawater, as well as by biological processes. "The daily peak concentration of silicate appeared almost at the daily lowest salinity" (Page 5 Line 25) indicates groundwater is also the source of silicate. Groundwater discharge varies with tides and is enriched in nitrate, phosphate and silicate, while depleted in nitrite in this area. So besides biological contribution, nitrite is sourced from offshore seawater, while nitrate, phosphate and silicate are sourced from groundwater.

P7, line 2, compared with "offshore water". You mean offshore seawater?

Response: yes. In the revision "water" is changed to "seawater" (Page 7 Line 6).

The authors should pay much attention to the use of significant digit. For example, "The NOx:P ratio varied from 4.78 to 12.94 in the spring-neap tide".

Response: revisions are made (Page 1 Line 10-11, 23-24; Page 5 Line 22, 28-31; Page 7 Line 28; Page 9 Line 25; Page 10 Line 6, 9).

Figure 1, the color scale is not clear. In addition, it is better to show water depth and current.

Response: the color scale is changed and bathymetry is added. No current data is available for this season for Sanya Bay.

[revised manuscript text omitted]

---

## Author Response (AR3)

Responses are in blue with page and line numbers where changes are made in the revision.

Review of the manuscript entitled "Tidal variability of nutrients in a coastal coral reef system influenced by groundwater" by Wang et al. (BG 2017 - 156)

Nitrification has been considered to be a two-step process consisting of ammonia oxidation (ammonium to nitrite) followed by nitrite oxidation (nitrite to nitrate), and each step is carried out by different microbes. Not much data are available on the rates of nutrient uptake at the low concentrations of nutrients occurring naturally in reef waters. Predictions of sustainable uptake/release rates in the natural environment based on uptake from enriched seawater with submarine groundwater discharge are questionable without detailed measurements and models. In this manuscript measurement of nitrite, nitrate without ammonia are reported which will not provide a complete picture in the absence of microbial processes that controls the DIN make up in estuarine/coral system. Corals may be capable of adaptive changes in uptake kinetics dependent on nutrient availability. However, the rate of nitrogen acquisition appears to be influenced on a diel cycle, presumably due to depletion of photosynthetic products during the night. Towards this, the manuscript is not strong in presenting the results and conclusions. It is, however, necessary to acknowledge the controversy and to address the potential biases associated with the choices made in the calculations.

Response: the controversy and potential biases are added in the discussion in the revision (Page 9 Line 22, Page 10 Line 11).

Almost 80% of SGD, calculated using the salinity difference between time-series station CT and the close seawater station, contributes very little increase nutrient concentrations and attributed due to biological processes as mentioned in the text. This contradicts the sentence on Page 3 line 25. What is the concentration of silicate, phosphate and nitrogen species in the river water and groundwater? Including this information will improve the reliability of the data discussed.

Response: the information of river water and groundwater is added in the revision (Page 3 Line 26).

I believe this manuscript is an important contribution to the field, however, much editing is needed before publication.

The specific comments are listed below:

Page 4 line 11: Prove volume of seawater

Response: the volume of Ra samples is added in the revision (Page 4 Line 13).

Page 4 line 19: Too simplistic. Delete this sentence.

Response: the suggestion is taken.

Page 5 line 2:    Delete "using Microsoft Excel (2007)"

Response: the suggestion is taken.

Page 5 line 28: Check the values and correct the sentence.

Response: we check the values. They are correct. The numbers are diurnal variations in $^{228}$Ra, which are the differences between the maximum and the minimum activities in each day, not the activity of $^{228}$Ra varying in each day.

Page 6 line 23: spelling! Write 'vertical' instead of 'verticle'

Response: thanks for catching this. Revision is made (Page 6 Line 23).

Page 8 line 3: Therefore we propose…. Not clear. Rewrite the sentence.

Response: the sentence is revised (Page 8 Line 3).

Page 8 line 15: During the flood tide….    This sentence is not clear. Rewrite it.

Response: the sentence is revised (Page 8 Line 15).

Page 8 line 20: The slope and constant values quoted in the equations 2 to 3 of section 4.2 are with more than significant digits of the salinity and nutrient concentrations used for getting them. Also you should include error on these values to show whether they are significantly different beyond the precision of measurement of the nutrients. Also correct them in Fig. 8. Delta P bio values are especially not different from the precision of measurement.

Response: revisions are made to keep 2 significant numbers with errors included (Page 8 Line 21-23, Fig. 8).